# Dual human lung models reveal compartment-specific activity of anti-tuberculosis drugs and host-directed therapies

Caio César Barbosa Bomfim,[1] Natacha Faivre,[1,2] Thomas Benoist,[1] Manon Popis,[1] Bastien Suire,[1] David Pericat,[1] José Manuel Sánchez-López,[1,3,4] Beatriz Melissa Aponte-Castillo,[3] Emmanuelle Näser,[1] Pénélope Viana,[1] Nicolas Guibert,[5] Romain Vergé,[1,6] Julien Mazières,[5] Arnaud Métais,[1] Renaud Poincloux,[1] Brigitte Raynaud-Messina,[1] Fabrice Dumas,[1] Olivier Neyrolles,[1,2] Christel Vérollet,[1,2] Etienne Meunier,[1] Geanncarlo Lugo-Villarino,[1,2] Céline Cougoule[1]

**ABSTRACT**   Tuberculosis (TB) remains a major global health challenge that requires new therapeutic strategies to improve drug efficacy, shorten treatment duration, prevent drug resistance, and limit *Mycobacterium tuberculosis* (Mtb) persistence. Here, we established complementary *in vitro* human lung models integrating alveolar macrophage-like (AML) cells and airway air–liquid interface (ALI) cultures to evaluate standard-of-care antibiotics, host-directed therapies, and virulence-targeting agents. AMLs recapitulated key morphological, transcriptional, and functional features of primary alveolar macrophages, including a CD16+ immunoregulatory phenotype highly permissive to Mtb infection. In parallel, ALI cultures maintained epithelial barrier integrity and secretory functions, allowing apical Mtb infection, drug penetration analysis, and inflammatory profiling. Benchmarking of standard-of-care antibiotics revealed compartment-specific activity: isoniazid, rifampicin, and moxifloxacin were effective in both systems, while pyrazinamide was active only in AMLs. Anti-inflammatory host-directed therapies, such as ibuprofen and doramapimod, selectively reduced cytokine production without affecting bacterial load. Together, this dual-platform system offers a physiologically relevant and scalable model to assess antimicrobial efficacy and host modulation across distinct pulmonary niches, bridging the gap between conventional macrophage assays and the complex human lung.

**IMPORTANCE**   Tuberculosis remains one of the world's deadliest infectious diseases. The development of new therapies is limited by the absence of human-relevant models that reproduce the distinct lung niches encountered by *Mycobacterium tuberculosis*. Current macrophage or epithelial monocultures fail to predict how drugs act in the alveolar versus airway compartments, where intracellular and extracellular bacteria coexist and trigger different immune responses. Here, we introduce a dual human lung platform integrating alveolar macrophage-like cells and air–liquid interface airway epithelium. These models recapitulate key physiological features, including macrophage immunoregulatory programming, epithelial barrier function, mucociliary activity, and compartment-specific drug penetration. Benchmarking standard antibiotics, host-directed therapies, and antivirulence strategies revealed striking niche-dependent differences in antimicrobial and immunomodulatory activities. This system provides a powerful and accessible preclinical tool to evaluate antimicrobial and host-directed interventions in relevant human lung environments, helping bridge the gap between simplified *in vitro* assays and the complex biology of human tuberculosis.

**KEYWORDS**   *Mycobacterium tuberculosis*, alveolar macrophages, air–liquid interface cultures, host-directed therapy, antibiotics, antimicrobial efficacy

**Peer Reviewer** Susanta Pahari, Texas Biomedical Research Institute, San Antonio, Texas, USA

Address correspondence to Caio César Barbosa Bomfim, Caio.Bomfim@ipbs.fr, or Céline Cougoule, Celine.Cougoule@ipbs.fr.

Caio César Barbosa Bomfim and Natacha Faivre contributed equally to this article. Author order was determined based on relative seniority.

Geanncarlo Lugo-Villarino and Céline Cougoule contributed equally to this article.

The authors declare no conflict of interest.

See the funding table on p. 22.

Tuberculosis (TB) remains a major global public health challenge, affecting millions of people every year. Caused by *Mycobacterium tuberculosis* (Mtb), it accounts for over 10 million new infections annually and continues to be the leading cause of death from a single agent, claiming approximately 1.25 million lives lost in 2023 alone (1). Although several effective antibiotic regimens exist, TB still shows a high relapse rate, with treatment success achieved in about 88% of patients with drug-sensitive TB (1, 2). Standard treatment for drug-sensitive TB lasts approximately six months. However, the prolonged duration of therapy often leads to poor patient adherence, contributing to relapse and the emergence of drug resistance. In cases of rifampicin-resistant or multidrug-resistant TB, treatment regimens are even longer, lasting up to 24 months, and are associated with lower treatment success rates (approximately 68%), further exacerbating the global disease burden. Together, these challenges underscore the urgent need for new and more effective treatment options.

Mtb has adapted to evade the host immune response, especially by interfering with macrophage microbicidal functions. These mononuclear phagocytes play a unique role in TB, serving as both the primary reservoir for Mtb replication and the key effector cells that control the infection. This dual function is central to the outcome of TB disease. Importantly, although many individuals encounter the bacteria, only about 5% to 10% develop active TB (3), which highlights the effectiveness of innate and adaptive immune responses in conferring host resistance (4). This has sparked interest in therapeutic strategies that target the interaction between immune cells and Mtb. By boosting macrophage ability to fight bacteria or countering the immune suppression caused by Mtb, for example, these so-called host-directed therapies (HDTs) may enhance the effectiveness of antibiotics. Consequently, this may lead to shorter treatment times, lower relapse rates, and help prevent the emergence of MDR Mtb strains.

Recent research reveals that a macrophage's origin and its specific tissular niche shape its phenotypic and functional characteristics (5). The lung niche comprises unique local signals, such as growth factors and metabolic cues, along with interactions with surrounding tissue components. Understanding these dynamics is particularly important in the TB context, as ontogeny and the specific lung niche influence how macrophages respond to and control Mtb (6, 7). Therefore, for therapeutic approaches targeting the interaction between macrophages and Mtb to be effective, it is essential to consider both their origin and the specific lung niche in which they reside.

Mtb primarily infects alveolar macrophages (AMs) upon initial lung invasion, exploiting their supportive metabolic environment and immune tolerance (6, 8). These macrophages are long-lived and self-renewing, mainly originating from the yolk sac and fetal liver during development. They are replenished by blood circulating monocytes derived from the bone marrow, particularly in older individuals or following an inflammatory insult, such as an infection (5, 9). Within the alveolar niche, AMs reside in elevated oxygen levels and lipid-rich surfactant, as well as in signals from airway epithelial-derived cytokines, such as granulocyte-macrophage colony-stimulating factor (GM-CSF) and transforming growth factor-β (TGF-β) (10). These environmental signals promote a tolerogenic phenotype in AMs by inhibiting the activation of pro-inflammatory pathways, which is essential for maintaining pulmonary homeostasis and preventing unnecessary inflammation against inert or non-pathogenic particles. However, this also creates an intracellular environment that is permissive to bacterial replication (6, 7, 11). Understanding the interplay between AMs and Mtb is crucial for proposing novel therapies that direct the host response.

As of today, most *in vitro* studies evaluating anti-TB drugs rely on macrophage cell lines, such as the human THP-1-based cellular models, or on M-CSF-driven human blood monocyte-derived macrophages (MDMs) (12–14). Yet, these models do not fully recapitulate the characteristics and functions of AMs, nor do they adequately represent the alveolar niche. Recent developments have introduced novel human AM models generated *in vitro* from pluripotent stem cells (15, 16) or from peripheral blood monocytes (17, 18), which offer greater experimental tractability. In particular,

the AM-like (AML) model adapted by Pahari and colleagues, using human monocytes differentiated with a specific combination of GM-CSF, TGF-β, interleukin 10 (IL-10), and surfactant, closely resembles primary AMs in terms of phenotype, metabolic activity, and functions. This model shows increased expression of peroxisome proliferator-activated receptor gamma (PPAR-γ) and greater susceptibility to intracellular Mtb proliferation compared to traditional MDMs (18). Likewise, recent advances in air–liquid interface (ALI) cultures derived from human bronchial epithelial cells (HBECs) provide key structural and functional features of the airway epithelium, delivering better reproducibility and suitability for medium- and high-throughput applications (19–21). Notably, these ALI cultures enable infection via apical inoculation of Mtb, mimicking the natural infection route, which leads to a fully developed infection and relevant physiological responses (22, 23). Nonetheless, the extent to which these *in vitro* approaches can reliably predict *in vivo* outcomes in human TB, particularly concerning host-targeted therapies, remains largely unexplored.

To address this, the present study established a dual *in vitro* system comprising the AML model and airway ALI cultures. We hypothesized that combining AML and ALI models would enable a more comprehensive and physiologically relevant assessment of both host-mediated and pathogen-directed determinants of Mtb infection and drug response, thereby increasing the translational relevance of preclinical findings to human TB. This system recapitulates the two main lung compartments engaged by Mtb: the intracellular AM niche, considered to be permissive for bacterial growth, and the airway barrier, which both restricts pathogen invasion and drives inflammation. Within this framework, the two models are designed to be complementary, with the AML system enabling detailed interrogation of macrophage heterogeneity, immune responses, and host–pathogen interactions at the cellular level, while the ALI cultures provide a physiologically relevant airway epithelial context that actively responds to infection by inducing chemotactic signals and antimicrobial defense programs that can also contribute to pathogen control. This combined approach enables the systematic evaluation of anti-TB compound testing, benchmarking their performance against molecules representing distinct pharmacological classes, including standard-of-care antibiotics, host-directed therapies, and virulence-targeting therapies. In this manner, this dual *in vitro* system aligns with the growing priority of co-targeting Mtb and the host to overcome therapeutic limitations.

## MATERIALS AND METHODS

### Bacterial culture

The genetically modified H37Rv-GFP and H37Rv-dsRED strains were grown in Middlebrook 7H9 medium supplemented with 0.05% Tween 80 (Euromedex) and 10% ADC (albumin, dextrose, and catalase; BD). The strain H37Rv::lux was constructed by transforming the Mtb wild-type strain H37Rv with the integrative plasmid pMV30-hsp-Lux13 (Addgene #26161) and cultivated with the same medium described before, supplemented with kanamycin (50 µg/mL). Both bacterial strains were maintained in culture at 37°C until the mid-log phase (OD 0.6–0.9) on day 7.

### Drugs

The benchmarking compounds used in this study were kindly provided by GSK Global Health Medicines R&D, including standard-of-care antibiotics, host-directed therapies, and virulence-targeting therapies. The identity, mechanism of action, and therapeutic class of each drug are summarized in Table S1, which provides annotated information relevant to TB control.

## AML and MDM generation, infection, and treatment

Human primary monocytes were isolated from the healthy subject's buffy coat (provided by Etablissement Français du Sang, Toulouse, France, under contract 21RB2025-028-R and 21RB2025-031-R). According to articles L12434 and R124361 of the French Public Health Code, the contract was approved by the French Ministry of Science and Technology (agreement number AC2009921). Written informed consents were obtained from the donors before sample collection. Monocytes (CD14+) were isolated from peripheral blood cells using magnetic positive selection (Miltenyi Biotec). For differentiation into MDM, the CD14+ monocytes were cultivated in a 24-well plate ($0.5 \times 10^6$ cells/well) for 7 days in RPMI-1640 Glutamax (Gibco), 10% fetal bovine serum (FBS, Sigma-Aldrich), 1% penicillin-streptomycin (Gibco), and human M-CSF (20 ng/mL, Miltenyi), as previously described (24).

For differentiation into AML, the CD14+ monocytes were cultivated in a 24-well plate ultra-low attachment ($2 \times 10^6$ cells/well) for 6–7 days with RPMI-1640 Glutamax medium supplemented with 10% FBS, 1% penicillin-streptomycin (Gibco), and IL-10 (5 ng/mL, Miltenyi Biotec), TGF-β (5 ng/mL, Miltenyi Biotec), and GM-CSF (10 ng/mL, Miltenyi Biotec), as well as Curosurf (100 μg/mL, Chiesi) (17, 18). The medium was refreshed every 2–3 days. After differentiation, the AMLs were infected with the H37Rv-GFP, H37Rv-DsRED, or H37Rv::lux Mtb strain (MOI indicated in the related figures) for 3 h (or overnight for the TEM experiment). Next, cells were washed three times with PBS to remove extracellular bacteria, plated ($2 \times 10^5$ cells/well) in a 96-well ultra-low-attachment plate, and then immediately treated after washing with different compounds (concentrations indicated in the related figures). Samples were incubated at 37°C and 5% $CO_2$ for 6 days.

## Bronchoalveolar lavage from patients

Bronchoalveolar lavage was performed using a flexible bronchoscope inserted through the nose with local anesthesia. The lavage samples were processed as outlined by (25), and the cells were stained for flow cytometry analysis. Patient samples were obtained through a contract between IPBS/CNRS and Toulouse Hospital (reference 257619, 19 April 2022), after informed consent, in accordance with the Declaration of Helsinki. Clinical and biological annotations of the samples have been declared to the CNIL ("Comité National Informatique et Libertés").

## Airway organoids

The organoids were derived from tumor-adjacent normal lung tissue obtained from patients with lung cancer undergoing surgical resection, as previously described (26, 27). Briefly, the tissue was dissociated into single cells and cultured in an extracellular matrix (Cultrex Basement Membrane Extract, Type 2, R&D Systems) with culture medium supplemented with various growth factors and inhibitors for 4 weeks, as originally described (27). The medium was refreshed every 3–4 days.

## Air–liquid interface (ALI) culture, infection, and treatment

Airway organoids were dissociated and seeded onto Transwell plate inserts, as previously described (28). The cultures were incubated submerged in PneumaCult Ex-Plus medium (StemCell Technologies) until reaching confluency. Once confluent, PneumaCult ALI medium (StemCell Technologies) was added exclusively to the basal side, leaving the apical side exposed to air (ALI). The cultures were then maintained at ALI conditions for an additional 3 weeks, with the medium refreshed every 2–3 days. For infection, Mtb ($4 \times 10^5$ bacilli in 20 μL per well) was added to the apical side of the ALI culture, with treatment administered in the basal medium. Cultures were incubated at 37°C for 6 days, with medium and treatment refreshed on day 3.

## Cytokine quantification

After 6 days of infection, the AMLs were centrifuged, and the supernatants were collected for analysis of cytokine production using Lumit immunoassay kits (Promega). For the ALI cultures, the medium from the basal side was harvested on days 3 and 6 post-infection, and cytokine levels were quantified by ELISA (Invitrogen).

## Bacterial load measurement

To quantify bacterial load by colony-forming unit (CFU) assay, the samples were lysed with 0.1% Triton X-100 (Euromedex) in water. Serial dilutions were prepared and plated on 7H11 agar medium supplemented with OADC. Colony counts were performed after 23 days of incubation at 37°C. In experiments where AMLs were infected with the H37Rv-GFP strain, the samples were fixed with PFA 4%, and bacterial load was assessed by measuring GFP fluorescence intensity using spectral flow cytometry (Northern Lights, Cytek). In experiments with AMLs infected with the H37Rv::lux strain, bacterial load was directly measured as relative light units (RLU) using the GlowMax luminometer (Promega). For ALI cultures, the tissue was lysed by adding 200 µL of lysis buffer (0.1% Triton X-100) to the inserts. The lysate was then collected, and the luminescence was measured using the GlowMax luminometer.

## Measurement of gene expression by RT-PCR

AML and MDM cell pellets were homogenized with 350 µL of RNA extraction buffer (RLT lysis buffer, Qiagen) by pipetting ten times, then briefly centrifuged at $2,000 \times g$ for 5 s. Homogenized cell lysates were then conserved at −80°C. RNA was extracted from cell lysates using a Qiagen RNeasy Mini Kit according to the manufacturer's instructions. RNA quality was assessed with the two measuring absorbance ratios Absλ = 260 nm/Absλ = 280 nm and Absλ = 260 nm/Absλ = 230 nm using a Thermo Fisher Nanodrop. The RNA was then stored at −80°C. For each sample, a fixed quantity (150–25 0ng) of extracted mRNA was reverse-transcribed into cDNA by mixing 1 µL of it with reverse transcriptase, deoxynucleotide triphosphates (dNTPs), oligo dTs, reverse transcription (RT) buffer solutions, and RNase-free water (for a 20 µL reaction), as part of the Verso cDNA Synthesis Kit. cDNA was stored at −20°C. A no-reverse-transcriptase control was also performed to test for remaining contaminating genomic DNA in the extracted RNA. The mixes were then heated in a Nexus GX2 Mastercycler thermocycler for the reaction to occur (30 min at 42°C followed by 2 min at 90°C). Finally, cDNA was mixed with forward and reverse primers for the genes of interest, along with SYBR Green Master Mix, in technical triplicate in a 96-well plate, and then subjected to RT-qPCR. Plates were loaded into a 7500 Real-Time PCR System thermocycler, which ran the following program: 50°C for 2 min, then 95°C for 10 min, followed by 40 cycles of 95°C for 15 s and 60°C for 1 min. The primer sequences used for this analysis are listed in the Supplementary Material (Table S2).

## Western blot

Cell lysates were homogenized by adding 1× Laemmli Sample Buffer (Bio-Rad) (95°C) and pipetting up and down ten times. Next, the samples were heated at 95°C for 5 min and stored at −20°C. Twenty microliters of sample and PageRuler Prestrained Protein Ladder (Thermo Fisher) were loaded onto 10% acrylamide gels, run at 120 V for 1 h, and then transferred onto polyvinylidene fluoride (PVDF) membranes. After transfer, the membrane was saturated for 1 h at room temperature in TBS-T (Tris 10 mM, pH 8, NaCl 150 mM, Tween 20 at 0.05%) containing 5% bovine serum albumin (BSA, Euromedex). Then, the membrane was incubated overnight at 4°C with the primary antibody under agitation. After three 10-min washes with TBS-T, the membrane was incubated with the secondary antibody conjugated to horseradish peroxidase enzyme (HRP) for 1 h at room temperature under agitation. Then, the membranes were washed three times for 20 min with TBS-T. The membranes were revealed using the WesternBright Quantum Western

blotting revelation kit (Thermo Fisher), and a WesternSure Pen (LicorBio) was used for the scale ladder. Images were acquired using the C-DiGit Blot Scanner (LicorBio). The primary and secondary antibodies are listed in the Supplementary Material (Table S3). Membranes could then be incubated in Antibody Stripping Buffer (1×) (Euromedex) for 5 min and washed three to five times in TBS-T, and then rinsed with distilled water to be reused for staining.

## Spectral flow cytometry

After infection at the indicated times, the culture medium was discarded, and Mtb-infected cells were collected, as previously described (24). Cells were then incubated with Human TruStain FcX (BioLegend) in a staining buffer (PBS, 1% FBS) for 10 min and labeled using ViaDye Red (Cytek, 1:2,000) in PBS (Gibco) at room temperature for 20 min. Then, cell surface proteins were stained for 20 min in a staining buffer containing True-Stain Monocyte Blocker (BioLegend) at 4°C. The fluorescent antibodies are listed in the Supplementary Material (Table S4). Cells were then fixed with 3.7% paraformaldehyde (PFA; EMS, DeltaMicroscopies) for 2 h and then resuspended in staining buffer. The samples were acquired using spectral flow cytometry with a Northern Lights (Cytek) instrument and analyzed with SpectroFlo V3.3.0 Software to subtract autofluorescence, followed by analysis with FlowJo V10 Software. The samples were cleaned and standardized using PeacoQC and the Mutual Nearest Neighbor method, respectively. For unbiased analysis, we used t-SNE on the concatenated sample (including all control and infected samples). Within the t-SNE map, one gate was created for the control group and another gate for the infected group.

## Scanning electron microscopy (SEM)

ALI culture was infected with H37Rv::lux (MOI 2:1) as described above. After 6 days, the culture medium was discarded, and cells were fixed with 3.7% PFA in PBS for 2 h. ALI was further fixed using 0.1 M sodium cacodylate buffer (pH 7.4) supplemented with 2.5% (vol/vol) glutaraldehyde (Euromedex). Then, they were washed three times for 5 min in 0.2 M cacodylate buffer (pH 7.4), post-fixed for 1 h in 1% (wt/vol) osmium tetroxide in 0.2 M cacodylate buffer (pH 7.4), and washed with distilled water. Samples were dehydrated through a graded ethanol series (25–100%), transferred to acetone, and critical-point dried with $CO_2$ using a Leica EM CPD300. Dried specimens were sputter-coated with 3 nm of platinum using a Leica EM MED020 evaporator and were examined with a FEI Quanta FEG250.

## Transmission electron microscopy (TEM)

AMLs were infected with Mtb-dsRED (overnight, MOI 2), as described above. After 24 h, the culture medium was discarded, and the cells were fixed using 3.7% PFA and 30 mM sucrose in PBS for 2 h. The cells were fixed in 2.5% glutaraldehyde and 2% PFA dissolved in 0.1 M Sorensen phosphate buffer (pH 7.2) (DeltaMicroscopies) for 2 h at room temperature, and then preserved in 1% PFA dissolved in 0.1 M Sorensen phosphate buffer. Adherent cells were treated for 1 h with 1% aqueous uranyl acetate, then dehydrated in a graded ethanol series and embedded in Epon. Sections were cut on a Leica Ultracut microtome, and ultrathin sections were mounted on 200-mesh Formvar- and carbon-coated copper grids. Finally, thin sections were stained with 1% uranyl acetate and lead citrate and examined using a transmission electron microscope (Jeol JEM-1400) at an accelerating voltage of 80 kV. Images were acquired using a digital camera (Gatan Orius) and colorized with the Photoshop software (Adobe).

## Histological staining

ALI cultures were fixed for 48 h in 10% neutral-buffered formalin (Sigma), rinsed thoroughly, resuspended in Hank's Balanced Salt Solution (HBSS, Gibco), and then stored at 4°C until further processing. After being embedded in paraffin, the sections (5 µm)

were stained with hematoxylin (BioGnost HEMML-OT-100) and eosin Y (Sigma). The cells were stained with Alcian Blue 8GX (Fluka 66011-500 mL-F) for 30 min, washed in tap water for 2 min, and counterstained with nuclear fast red (Sigma) for 5 min. Images were acquired using a Zeiss Axio Imager M2 with either a 10×/0.3 EC Plan-Neofluar or 20×/0.8 Plan Apochromat Zeiss objective, and a Zeiss Axiocam 503 color camera. Images were processed using the Zeiss Zen software.

## Bioinformatic analysis

For the bioinformatic analysis, two different transcriptome data sets were used. Counts and TPM (transcripts per million) matrices were obtained from the NCBI GEO (Gene Expression Omnibus) platform using the accession numbers GSE188945 and GSE189996. The transcriptome profiles were then compared in two scenarios: MDM versus AML, and Mtb-infected AM at 72 h versus uninfected AM. The counts matrix was used to perform the differential expression analysis (DEG) with the DESeq2 algorithm. The false discovery rate (FDR) was managed using the Benjamini-Hochberg method. Genes with an FDR-adjusted $P$-value of less than 0.05 and an LFC greater than 1.5 or less than −1.5 were deemed differentially expressed (29). Gene ontology (GO) terms and biological processes (BP) that were overrepresented were identified using the ClusterProfiler package, based on a false discovery rate below 0.05 and an enrichment value of 1 (30). The TPM matrix was utilized for data visualization. Box plots, volcano plots, and heatmaps were generated for the most representative genes of interest. All analyses and visualizations were performed using the R programming language (version 4.5.1) and RStudio, with the help of the ggplot2, DESeq2, and ComplexHeatmap packages.

## Statistical analysis

Graphs and statistical analyses were performed using Prism software (version 10.5 GraphPad). Non-parametric tests were applied, with paired Wilcoxon or unpaired Mann–Whitney tests used as indicated in each figure legend. Statistical significance was defined as a $P$-value of ≤0.05. Data are presented as mean ± SD. All statistical comparisons were based on pairwise comparisons between each drug-treated condition and the corresponding infected, untreated control.

## RESULTS

### Molecular profiling and phenotypic characterization of the human alveolar macrophage-like cell model

To dissect the unique interactions between Mtb and AMs in the pulmonary microenvironment (Fig. 1A, left), we used the recently described human monocyte-derived AML model, which recapitulates key features of primary human AMs (hAM) (11, 17, 18, 31). Briefly, human monocytes were differentiated for 6 days with a combination of GM-CSF, TGF-β, IL-10, and surfactant to obtain AMLs (Fig. 1A, right). First, we confirmed the phenotype of these AMLs by comparing them with MDMs. Consistent with previous studies (17, 18), AMLs exhibited higher gene expression typical of primary hAMs, such as *MRC1* (CD206), *MARCO*, *CXCL3*, *DUSP1*, *PPARG* (PPAR-γ), and *SPI1* (PU.1), relative to MDMs from the same donor (Fig. S1A). In contrast, MDMs expressed significantly higher levels of *MMP7* and *MMP9* compared to AMLs (Fig. S1A). At the protein level, AMLs showed higher PPAR-γ expression compared to MDMs (Fig. S1B), supporting the notion that this AML cell model relies on this transcription factor for its development and functions, as primary hAMs do (17, 18).

To determine whether AMLs resemble primary hAMs, we compared surface marker expression in hAMs isolated from bronchoalveolar lavage (BAL) samples of patients with various lung pathologies with AMLs. The majority of hAMs recovered from BAL consisted of a CD206$^+$CD64$^+$ cell population, within which most cells also expressed CD14 and CD16 (Fig. 1B; Fig. S1C). These results are consistent with the literature, which reports high expression of these markers in hAM (32–34). Interestingly, AMLs exhibited a similar

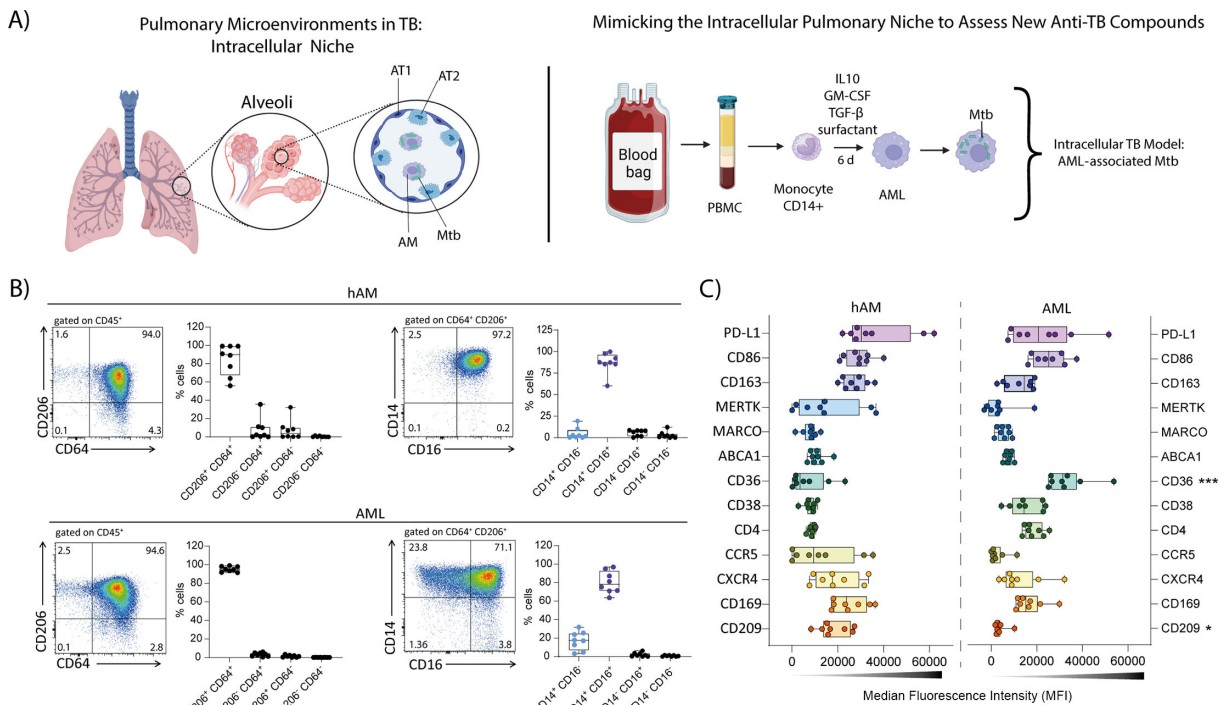

FIG 1  Phenotypic characterization of the human AML cell model reproduces the anti-inflammatory and immunoregulatory phenotype of hAMs. (A) Schematic illustration of alveolar macrophage-like cells (AML) generation from peripheral CD14$^+$ monocytes (created with BioRender.com). (B) Flow cytometry dot plots showing the expression of CD206 vs CD64 and CD16 vs CD14 in human alveolar macrophages (hAMs) recovered from bronchoalveolar lavage (BAL) and AMLs, and corresponding quantification of the indicated cell subsets. (C) Graphs showing the median fluorescence intensity (MFI) of different markers in the CD206$^+$CD64$^+$CD14$^+$CD16$^+$ subpopulation of hAMs (left) and the corresponding subpopulation of AMLs (right). In the graphs, each dot represents an individual donor. Significant differences were determined using an unpaired, non-parametric Kruskal–Wallis test. $P < 0.05$ (*), $P < 0.01$ (**), $P < 0.001$ (***).

phenotype, characterized by a high frequency of CD206$^+$CD64$^+$ macrophages, predominantly CD14$^+$CD16$^+$ within this gated population (Fig. 1B). Further analysis with a broader panel of cell-surface markers revealed that AMLs closely resemble the phenotype of hAMs. Various key markers, including PD-L1, CD86, CD163, MERTK, MARCO, CXCR4, and CD169, showed similar expression patterns between the two cell types, reinforcing AMLs as a physiologically relevant *in vitro* model for primary hAMs (Fig. 1C).

To complement this phenotypic characterization of AMLs, we leveraged a publicly available transcriptomic data set comparing AML cell model with hAMs and MDMs (17, 18). For this data set, we extracted a curated panel of genes, including those identified through our phenotypic analysis and reflecting macrophage polarization, lipid metabolism, scavenger receptor activity, matrix remodeling, cytokine production, and immune checkpoint regulation. Consistent with our *in vitro* phenotyping, AMLs displayed a transcriptional profile more closely aligned with anti-inflammatory and tissue-resident markers, including high expression of cell-surface marker genes, such as *FCGR1A* (CD64), *FCGR3A* (CD16), *CD163*, *MRC1* (CD206), and *MARCO*, as well as the transcription factors *PPARG* (PPAR-γ), similar to hAMs from healthy donors (Fig. S2A). These results are reinforced by a tendency toward a higher expression of *SIGLEC1* (CD169) and *CXCR4* in AMLs. In contrast, MDMs showed a marked increase in the matrix-remodeling–related gene *MMP9*, along with a trend toward higher expression of pro-inflammatory markers, including *TNF*, *CD86*, *CD38*, *CCR5*, *CD4*, and *FCAR*. Taken together, these results are consistent with MDM activation toward a more inflammatory, monocyte-derived state (Fig. S2A).

Beyond the predefined macrophage gene signature, we identified the top genes significantly enriched in AMLs compared with MDMs. The resulting volcano plot revealed a clear separation between these cell populations, as previously reported (17, 18).

Among the most highly upregulated genes in AMLs, several have documented roles in immune regulation, tissue remodeling, and metabolism (Fig. S2B). Notably, *ABHD5*, a lipase co-activator, has been shown to suppress NF-κB-dependent MMP expression (35), suggesting a potential link to the lower *MMP7/MMP9* levels observed in AMLs. Interestingly, transcription factor *KLF4*, known to integrate STAT6-dependent alternative activation pathways (36), was prominently upregulated, further supporting the tissue-resident, M2-skewed features of AMLs. Conversely, several genes were markedly downregulated in AMLs compared to MDMs. Among them, *GALM*, a galactose mutarotase involved in glucose metabolism, and *TRAF1*, a TNF receptor–associated factor with established roles in NF-κB signaling and inflammasome activation (37), may reflect metabolic and inflammatory programming differences between the two macrophage populations (Fig. S2B). Finally, gene ontology (GO) enrichment analysis of AML-upregulated genes highlighted biological processes consistent with their lung-adapted phenotype, including G protein–coupled receptor signaling, response to bacterium, blood vessel development, epithelial cell differentiation, regulation of inflammatory response, lipid homeostasis, and humoral immune response (Fig. S2C). These pathways further emphasize the specialized, tissue-resident programming of AMLs, which integrate metabolic adaptation, immune regulation, and barrier-associated functions.

Collectively, the AML model closely resembles the morphology, surface markers, and transcriptional profiles of primary hAMs and differs from those of MDMs. AMLs exhibit tissue-resident, anti-inflammatory, and lipid metabolic signatures, with reduced expression of matrix remodeling and pro-inflammatory genes. These characteristics align with the functions of the alveolar niche, establishing AMLs as a relevant and manageable surrogate for hAMs.

## Mtb infection reveals phenotypic heterogeneity and distinct activation states in the human alveolar macrophage-like cell model

To assess whether phagocytic function is preserved after differentiation into AML, we exposed AMLs to Mtb H37Rv expressing GFP. Cells were examined by epifluorescence and light microscopy, as well as transmission electron microscopy (TEM), confirming the presence of Mtb bacilli within intracellular compartments (Fig. S3A; Fig. 2A, pink). Of note, TEM analysis revealed AMLs as rounded to oval in shape, featuring irregular plasma membranes that extend into short pseudopodia, a cytoplasm containing many vacuoles and lipid bodies (possible signs of surfactant clearance), and a large nucleus, slightly off-center with a distinct nucleolus (Fig. 2A). In infected cells, bacilli were predominantly localized within membrane-bound vacuoles consistent with phagosomes. While flow cytometry analysis 6 days post-infection revealed a dramatic increase in cell death at an MOI of 5:1, an MOI of 1:1 yielded an intact cell viability with approximately 50% of infected cells (Fig. S3B and C). As expected, quantification of bacterial burden over time using different strains and methods (H37Rv::lux for luminescence and H37Rv-GFP for FACS, and both for CFU) showed that Mtb replicates within AMLs (Fig. S3D).

We next examined the gene expression of AML following infection with Mtb H37Rv-GFP. Notably, we observed a significant reduction of *PPARG* and an increase of *MMP7*, *MMP9*, and *TNF*, indicating a transcriptional reprogramming toward an inflammatory response and matrix-degrading phenotype in AML (Fig. 2B). To further characterize the impact of infection, we examined the phenotypic profile of AMLs following infection with Mtb H37Rv-GFP. This analysis showed a marked expansion of the CD14$^+$CD16$^-$ subset (Fig. 2C). Analysis of GFP expression revealed that the CD14$^+$CD16$^+$ AML subset was significantly more permissive to Mtb infection than its CD14$^+$CD16$^-$ counterpart (Fig. 2D). We reasoned that these differences in susceptibility could reflect distinct phenotypic programs. To explore this, we next performed a high-dimensional t-SNE analysis on Northern Light spectral flow cytometry data, incorporating CD45$^+$ AMLs from infected and uninfected conditions (Fig. 2E). To this end, samples were concatenated and embedded based on the expression of markers associated with pro-inflammatory activation (i.e., CD86, CD38, CD64) and inflammatory checkpoints (i.e., PD-L1, which,

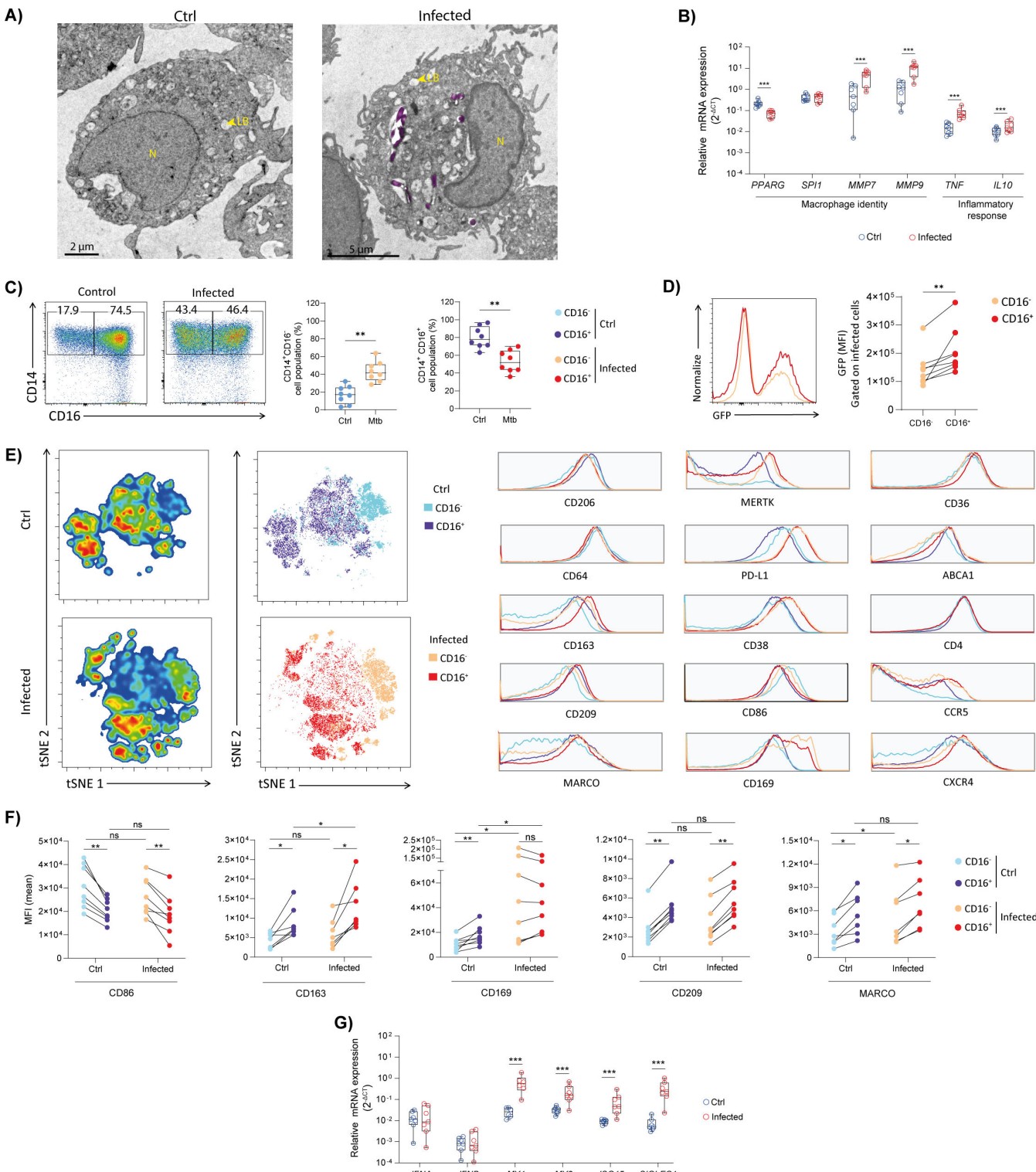

**FIG 2** Mtb infection induces a phenotypic heterogeneity within the AML compartment. AML cells were infected *in vitro* with H37Rv-GFP (MOI 1:1) for 4 h, washed, and then incubated for 6 days. (A) AML uninfected and Mtb-infected were visualized using transmission electron microscopy (TEM). Bacteria are shown in magenta. Scale bars: 2 µm. LB (lipid body); N (nucleus). (B) Relative gene expression of *PPARG*, *SPI1* (PU.1), *MMP7*, *MMP9*, *TNF,* and *IL10* in control (uninfected) and Mtb-infected AML cells assessed by RT-qPCR. (C) Flow cytometry dot plots and graphs showing the frequency of CD14⁺CD16⁻ and CD14⁺CD16⁺ subpopulations in control (uninfected) and Mtb-infected AML cells. (D) Flow cytometry histogram and graph showing the median fluorescence intensity (MFI) of Mtb in CD14⁺CD16⁻, and CD14⁺CD16⁺ subpopulations among Mtb-infected AML cells. (E) Unsupervised t-SNE analysis of flow cytometry data from control

Fig 2 (Continued)

(uninfected) and Mtb-infected AMLs, stratified by CD16 expression. Histograms show the MFI of different markers analyzed in CD16$^+$ and CD16$^-$ subpopulations. (F) Histograms and graphs showing the median fluorescence intensity (MFI) of CD86, CD163, CD169, CD209, and MARCO in CD14$^+$CD16$^+$ and CD14$^+$CD16$^-$ AML subpopulations in control (uninfected) and Mtb-infected AML cells. (G) Relative gene expression of *IFNA*, *IFNB*, *MX1*, *MX2*, *ISG15,* and *SIGLEC1* in control (uninfected) and Mtb-infected AML cells assessed by RT-qPCR. In the graphs, each dot represents an individual donor. Significant differences were determined using a paired, non-parametric Wilcoxon test. $P < 0.05$ (*), $P < 0.01$ (**), $P < 0.001$ (***); ns, not significant .

although immunoinhibitory, is strongly induced by inflammatory stimuli). We also included key regulatory and anti-inflammatory molecules (i.e., CD163, CD206, MERTK, MARCO), lipid metabolism, and scavenger receptors (i.e., ABCA1, CD36). This strategy enabled us to resolve infection-driven shifts in activation states and to provide a framework for linking infection status with functional phenotypes. Using this approach, AMLs were separated into control and infected populations, and within each group, we further stratified subsets based on CD16 expression (Fig. 2E). Comparative analysis of surface marker expression among these groups revealed various notable differences. As expected, PD-L1 was substantially upregulated in both CD16$^+$- and CD16$^-$-infected cells compared to uninfected controls. MERTK, CD169, CCR5, CD163, and CD38 expression was increased explicitly in infected cells, independent of CD16 expression. Notably, we observed reduced levels of ABCA1 in CD16$^-$ uninfected and Mtb-infected cells. Moreover, we showed that the CD16$^+$ subset, independent of the infection, is characterized by a reduction of the pro-inflammatory marker CD86 and an increase of CD209 compared to the CD16$^-$ counterpart (Fig. 2E). Together, these findings indicate that CD16$^+$ and CD16$^-$ AMLs adopt distinct activation states in response to Mtb, potentially underlying their differential susceptibility to infection.

To further characterize these subsets, supervised flow cytometry analysis revealed that under uninfected conditions, CD14$^+$CD16$^+$ cells expressed higher levels of CD163, CD169, CD209, and MARCO, consistent with an M2-like anti-inflammatory phenotype that may enhance their ability to interact with and internalize microorganisms (Fig. 2F; Fig. S3E). In contrast, the CD14$^+$CD16$^-$ subset displayed higher expression of CD86, a marker associated with an M1-like pro-inflammatory state. Following Mtb infection, these phenotypic profiles were largely preserved: CD16$^+$ cells continued to express higher levels of CD163, CD209, and MARCO, whereas CD16$^-$ cells maintained a more pro-inflammatory profile. The main exception was CD169, a type-I interferon (IFN-I)-inducible activation marker, which was upregulated in both subsets after infection, most notably in CD16$^-$ cells, leading to comparable expression levels between CD14$^+$CD16$^-$ and CD14$^+$CD16$^+$ subpopulations (Fig. 2F; Fig. S3E). Consistent with an activated IFN-I response, bulk mRNA analysis of Mtb-infected AMLs revealed increased expression of several IFN-I-inducible genes upon Mtb infection, including *MX1*, *MX2*, interferon-stimulated gene 15 (*ISG15),* and *SIGLEC1* (CD169) (Fig. 2G). Collectively, this difference in expression profiles highlights a functional dichotomy between these AML subsets, which likely shapes their contributions to Mtb infection.

To place these findings in a more physiological context, we performed a complementary analysis using publicly available transcriptomic data from freshly isolated hAMs from healthy donors, which were then infected *ex vivo* with Mtb (38). This data set tracked transcriptional dynamics over 72 h. In line with our observations, infected hAMs upregulated activation genes (e.g., *CD274* [PD-L1], *CD38*, *TNF*, *IL6*) and progressively downregulated the anti-inflammatory regulator *FCGR3A* (CD16) (Fig. 3A). Beyond this predefined macrophage gene signature, the broader transcriptomic landscape of infected hAMs revealed strong induction of chemokines, ISGs, and inflammatory mediators (Fig. 3A and B). The IFN-I axis was characterized by the induction of several ISGs, including *CXCL10* (Fig. 3B), a pro-inflammatory chemokine linked to tissue pathology (39), further reinforcing the inflammatory potential of infected AMs. Additional ISGs comprised the *MX1* and *MX2*, classically described as viral restriction factors but also relevant to bacterial infection dynamics, as well as *ISG15*, which regulates cellular metabolism and mitochondrial activity (40). Importantly,

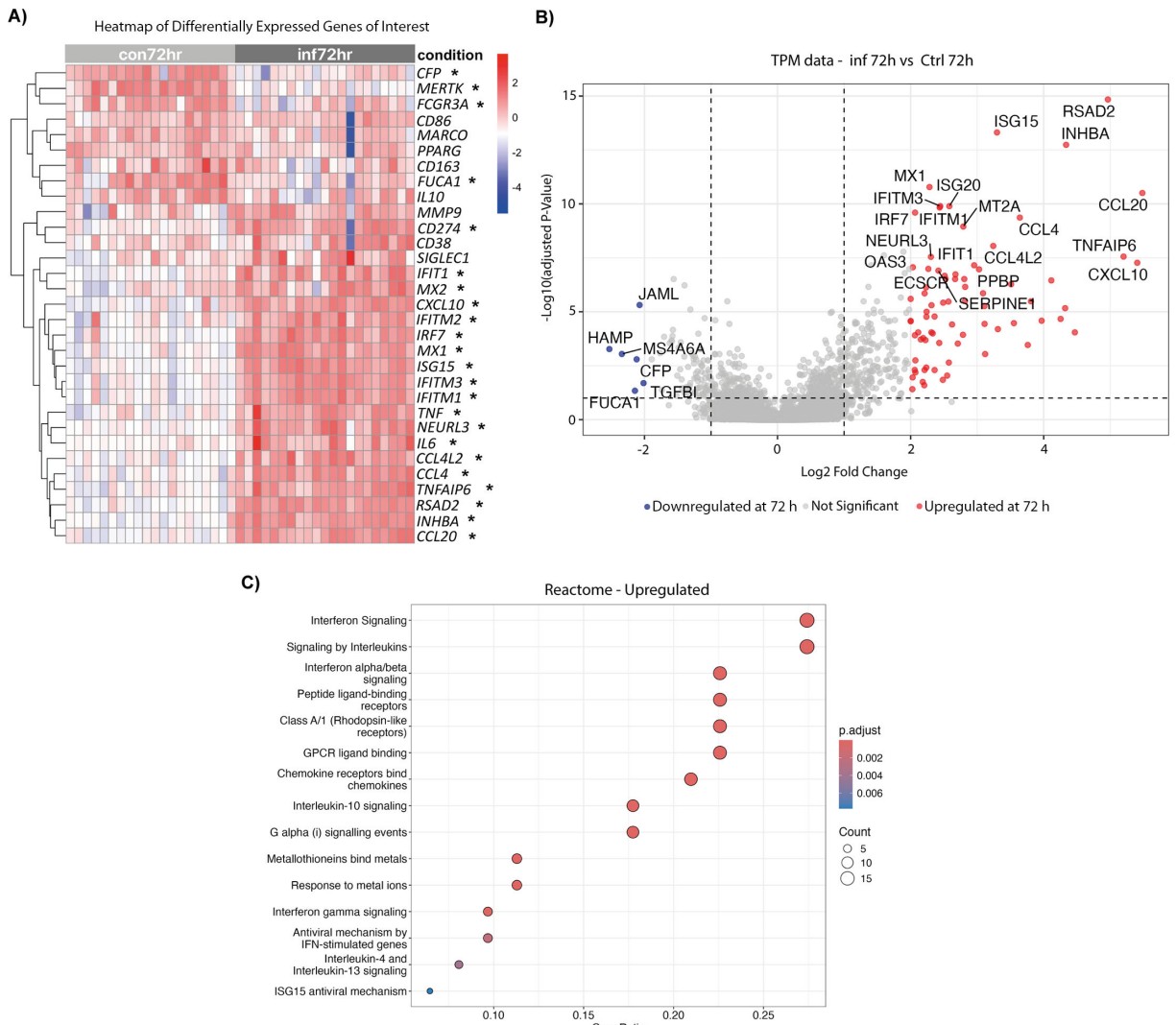

**FIG 3** Transcriptomic response of primary hAM to Mtb infection. Gene expression levels of primary hAM derived from different healthy donors and infected *ex vivo* with H37Rv for 72 h were analyzed using publicly available RNA-seq data sets. (A) Heatmap of selected differentially expressed genes between AML, hAM, and MDM cellular models. Red indicates upregulated genes, and blue indicates downregulated genes. Genes showing significant differences between AML and MDM (adjusted $P < 0.05$) are marked with an asterisk (*). (B) Volcano plot highlighting the top 40 DEGs (fold change $\geq \pm 1.5$; adjusted $P < 0.05$). Red dots indicate upregulated genes, blue dots indicate downregulated genes, and gray dots represent genes outside the threshold. (C) Reactome pathway enrichment analysis of upregulated genes in 72 h Mtb-infected hAMs compared with uninfected controls. Dot size reflects the number of genes per pathway, and color corresponds to the adjusted *P*-value.

transcriptional regulators such as *IRF7*, a master controller of IFN-I signaling implicated in M1-to-M2 macrophage transitions (41), and *NEURL3*, which promotes innate antiviral defense by catalyzing K63-linked ubiquitination of IRF7 (42), suggest a layered regulation of this IFN-I-driven program (Fig. 3A and B). Complementary reactome analysis revealed significant enrichment of IFN-I signaling, interleukin signaling, IL-10 and IL-4/IL-13 pathways, and chemokine receptor signaling in hAMs after Mtb infection (Fig. 3C).

Together, these data establish that Mtb-infected hAMs adopt a strong pro-inflammatory, IFN-I-dominated transcriptional profile, characterized by chemokine-driven leukocyte recruitment, antiviral-like defense programs, and the selective downregulation of complement and autophagy regulators. In line with our AML findings, these transcriptomic profiles underscore that Mtb infection drives dynamic reprogramming of AMs, balancing antimicrobial and inflammatory pressures with compensatory regulatory pathways that may ultimately contribute to bacterial persistence.

## Benchmarking of anti-TB drugs in the human alveolar macrophage-like cell model

To assess whether AMLs represent a relevant human *in vitro* platform for testing novel anti-TB compounds, we benchmarked their response to three drug categories: standard-of-care antibiotics, host-directed therapies, and virulence-targeting therapies. We first measured bacterial burden using luminescence readouts from AMLs infected with H37Rv::lux (Fig. 4A). Importantly, none of the compounds tested displayed detectable cytotoxicity at the concentrations used (Fig. S4), ensuring that the observed effects reflected genuine modulation of bacterial replication rather than host cell toxicity. As expected, most standard-of-care antibiotics markedly reduced luminescence, except pyrazinamide (PZA). Both virulence factors–targeting compounds achieved modest reductions in luminescence intensity (Fig. 4A). Among host-directed therapy candidates, only doramapimod (DORA (43)) demonstrated a strong inhibitory effect on luminescence. To validate these findings, we quantified viable bacteria using CFU counts, the gold-standard method for assessing Mtb replication (Fig. 4B). All standard-of-care antibiotics, including PZA, significantly suppressed bacterial replication. In addition, most host-directed therapies and virulence factor-targeting compounds reduced CFU counts. By contrast, ibuprofen (IBU), a non-steroidal anti-inflammatory drug (NSAID) (44), and, unexpectedly, DORA failed to yield statistically significant effects on CFU counts (Fig. 4B).

Beyond antibacterial activity, we evaluated the immunomodulatory potential of these compounds in AMLs following Mtb infection (Fig. 4C). To this end, we measured the secretion of pro-inflammatory cytokines TNF-α and IL-6, as they are recognized correlates of disease severity in TB patients (45). All standard-of-care antibiotics significantly reduced TNF-α production, but only linezolid (LZD) and rifampicin (RIF) also markedly

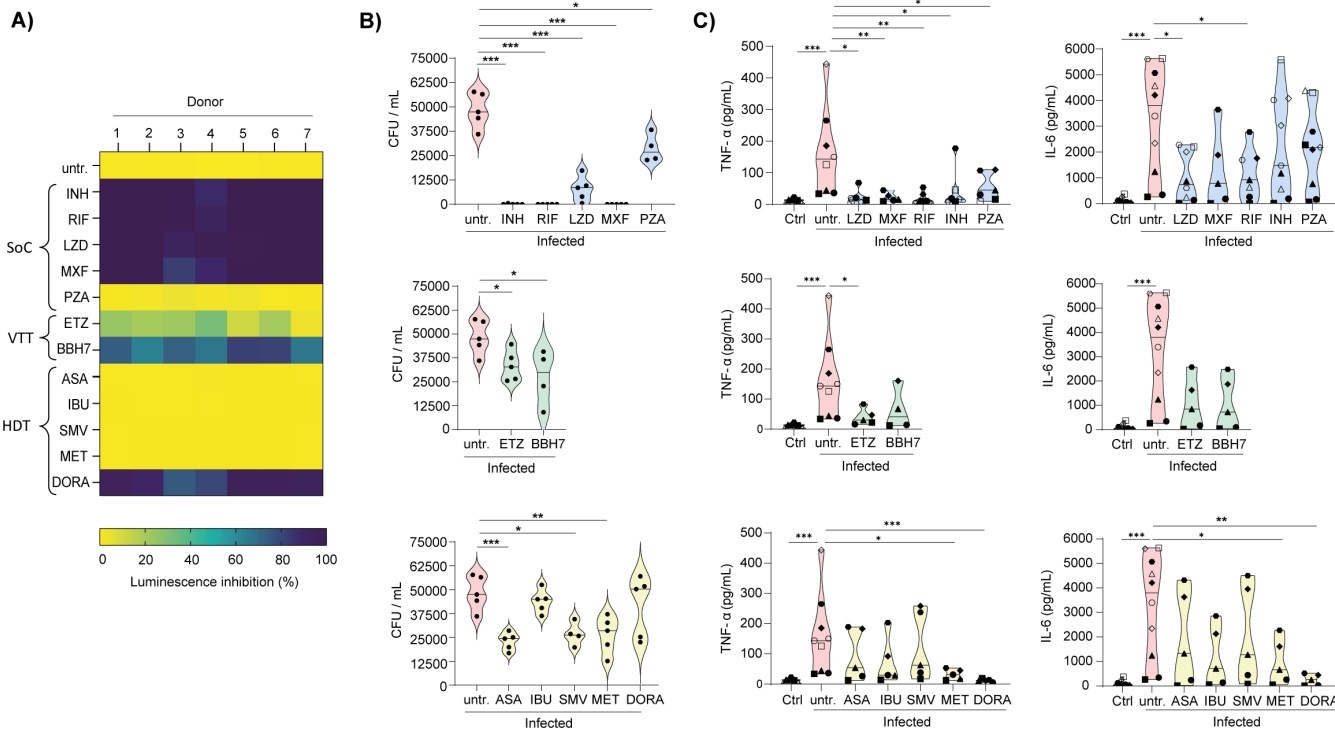

FIG 4 Evaluation of bacterial control and host response modulation by anti-TB agents in AML cells. AMLs were infected with H37Rv::lux at an MOI of 1:1 for 4 h, washed, and then treated with different compounds (SoC, standard-of-care antibiotics; VTT, virulence-targeting therapies; HDT, host-directed therapies), and incubated for 6 days. (A) Heatmap showing luminescence inhibition. (B) Colony-forming unit (CFU) of Mtb-infected AML cells. (C) TNF-α and IL-6 production by Mtb-infected AML cells. In the graphs, each dot represents an individual donor. Compound concentrations tested were as follows: SoC (INH = 10 µM; RIF = 10 µM; LZD = 10 µM; MXF = 10 µM; PZA = 25 µM), VTT (ETZ = 10 µM; BBH7 = 10 µM), HDT (ASA = 50 µM; IBU = 50 µM; SMV = 10 µM; MET = 50 µM; DORA = 50 µM). Significant differences were determined using an unpaired, non-parametric Mann–Whitney test. $P < 0.05$ (*), $P < 0.01$ (**), $P < 0.001$ (***).

decreased IL-6 levels (Fig. 4C). Among the virulence factors targeting compounds, ethoxzolamide (ETZ) uniquely suppressed TNF-α without affecting IL-6 (Fig. 4C). Within the host-directed therapy group, metformin (MET)—a biguanide anti-diabetic agent repurposed for TB research through activation of AMP-activated protein kinase (AMPK), thereby promoting autophagy and enhancing mitochondrial reactive oxygen species (mROS) production (46)—and DORA both significantly reduced TNF-α and IL-6 levels (Fig. 4C). Taken together, these findings reveal that the AML model can identify distinct immunomodulatory profiles across various classes of anti-TB compounds, reinforcing its significance as a physiologically relevant hAM platform and as a practical alternative to MDM- or THP-1-based approaches.

## Human airway air–liquid interface (ALI) system recapitulates the human pulmonary environment and supports Mtb infection

Mtb is widely recognized as an intracellular pathogen that primarily resides within macrophages. However, during active TB, a substantial proportion of the bacilli are present in the extracellular environment of the airways. In this milieu, extracellular Mtb directly engages the airway epithelium, triggering, in turn, pattern recognition receptor-mediated signaling cascades that culminate in the secretion of pro-inflammatory cytokines and chemokines (22, 47). This epithelial response not only amplifies local inflammation and modulates resident macrophages but also recruits and activates additional leukocytes, ultimately influencing the course of infection and incidence of tissue pathology.

To model the epithelial response to Mtb and its contribution to airway inflammation, we employed an airway organoid/ALI culture system, an *in vitro* approach that mimics the structure and function of bronchiolar epithelium (Fig. 5A) (26–28, 48). To confirm the model's relevance, we first examined whether it recapitulates key features of the airway epithelium. Epifluorescence and light microscopy analysis of the ALI cultures revealed a densely packed epithelial layer with a healthy tissue architecture. The presence of wave-like structures and surface texture variations may reflect the formation of cilia or microvilli, hallmark features of differentiated airway epithelial cells (Fig. 5B). Indeed, analysis of H&E staining demonstrated a thin, regular, and well-organized cell layer with a morphology typical of healthy airway tissue (Fig. 5C). The presence of nuclei at varying heights and structures suggestive of an apical ciliated border is consistent with a pseudostratified, ciliated epithelium (Fig. 5C). Additionally, Alcian Blue staining demonstrated secretory cells distributed throughout the epithelial layer, together with apical acidic mucins and apical mucus accumulation, indicative of mucosecretory differentiation (Fig. 5D). To complement these findings, we performed SEM to visualize the apical architecture of the ALI cultures directly. The surface ultrastructure confirmed the presence of ciliated epithelial cells with numerous surface cilia, scattered microvilli, and goblet cells, supporting the differentiated state of the ALI cultures (Fig. 5E). Assessment of epithelial integrity and paracellular permeability demonstrated that ALI cultures form a functional barrier, as indicated by low dextran flux and high transepithelial electrical resistance values (Fig. 5F and G). Finally, live imaging using epifluorescence and light microscopy revealed continuous, synchronized ciliary beating, a key physiological mechanism of mucociliary clearance, as shown in Movie S1. These observations indicate that the ALI cultures not only exhibit structural features of airway epithelium but also preserve essential functional properties.

Next, we examined how ALI cultures respond to Mtb infection. To mimic the physiological route of airborne exposure, Mtb was applied exclusively to the apical surface of differentiated ALI cultures (22, 23), and mycobacterial replication was monitored over 6 days by measuring bioluminescence and live bacterial load (CFU). Strikingly, rather than replicating, Mtb showed a progressive decline over time in both luminescence and viable counts, indicating not only a lack of replication but also a potential loss of bacterial viability in this model (Fig. 5H), as previously described in Mtb-infected airway organoids (26). No bacteria were detected in the basal compartment at any time point, indicating

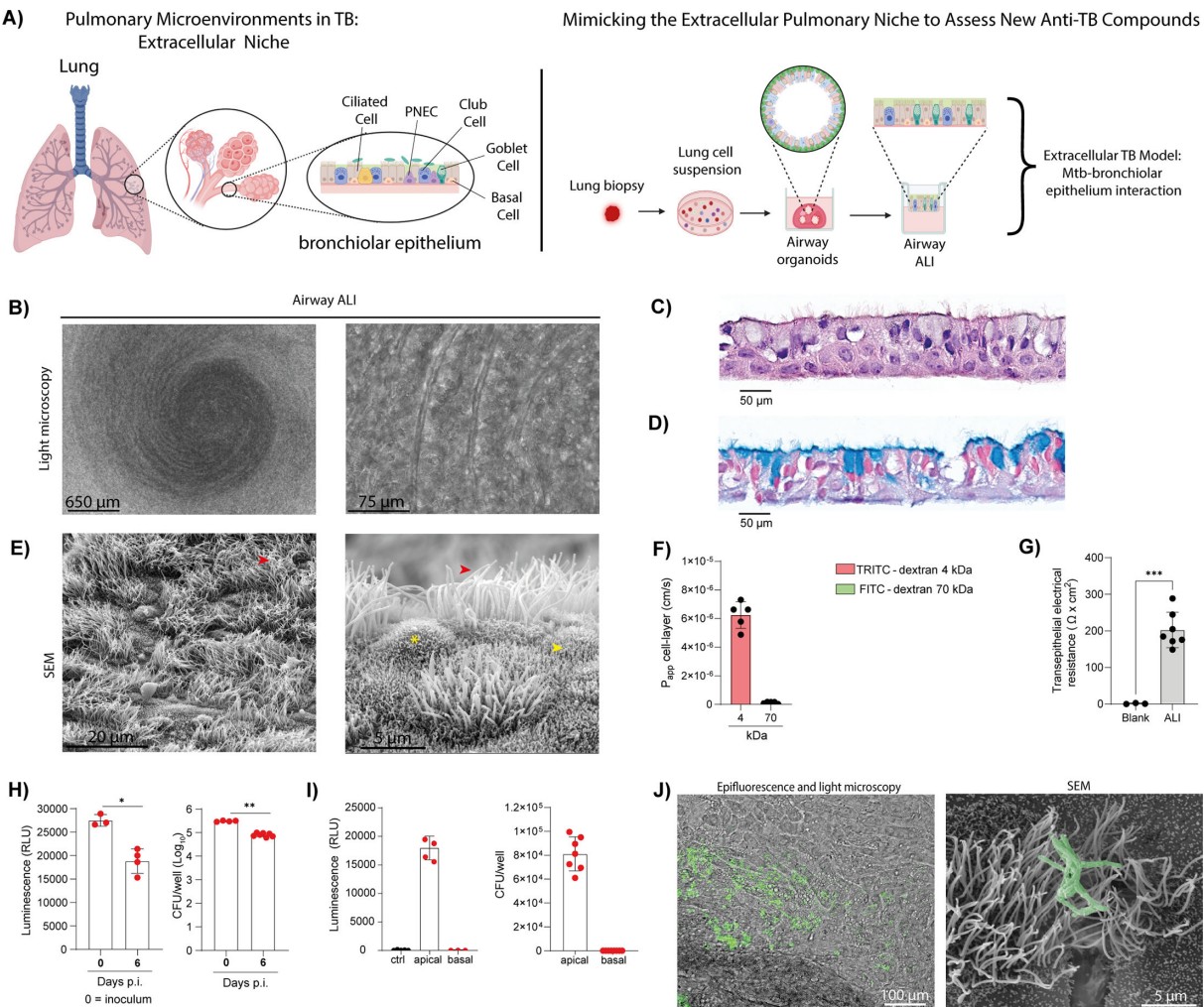

FIG 5 Airway ALI cultures recapitulate the human pulmonary environment and support Mtb infection. Human airway organoid/ALI cultures were derived from tumor-adjacent normal lung tissue obtained from different patients. For infection experiments (panels H–J), differentiated airway ALI cultures were infected with H37Rv::lux ($4 \times 10^5$ bacilli/insert) on the apical side and incubated for 6 days. (A) Schematic illustration of airway ALI culture generation from lung biopsy (created with BioRender.com). (B) Airway ALI cultures visualized using light microscopy. Scale bars: 650 µm or 75 µm. (C) Hematoxylin and eosin (H&E) staining on an airway ALI cross-section. Scale bars: 50 µm. (D) Alcian Blue staining of airway ALI cross-section. Scale bars: 50 µm. (E) Scanning electron microscopy (SEM) of airway ALI cultures showing a differentiated bronchial epithelium. Red arrows indicate cilia, asterisks indicate goblet cells, and yellow arrows indicate microvilli. Scale bars: 20 µm or 5 µm. (F) Dextran flux assay showing permeability defined by apical-to-basolateral flux (Papp) of TRITC-dextran 4 kDa or FITC-dextran 70 kDa. (G) Transepithelial electrical resistance measurement of airway ALI cultures. (H) Luminescence and CFUs measured in the apical compartment at day 0 and day 6 post-infection (p.i.). (I) Measurement of luminescence and CFUs at apical and basal compartments at day 6 p.i. (J) Scanning electron microscopy (SEM) image of Mtb-infected airway ALI cultures. Scale bar: 100 µm or 5 µM. Significant differences were determined using an unpaired, non-parametric Mann–Whitney test. $P < 0.05$ (*), $P < 0.01$ (**), $P < 0.001$ (***).

that Mtb did not alter the epithelial barrier and remained confined to the apical side (Fig. 5I). Epifluorescence and light microscopy, as well as SEM, confirmed this observation and further revealed intimate interactions between Mtb and the cilia of ciliated epithelial cells at the apical surface (Fig. 5J). Collectively, these findings demonstrate that while ALI cultures allow Mtb to colonize the apical airway, they still function as a barrier against bacterial translocation and deeper invasion, recapitulating a key aspect of the airway's innate defense. Thus, in contrast to the permissive and heterogeneous intracellular environment observed in AMLs, the ALI model represents a restrictive epithelial niche in which Mtb fails to replicate extracellularly.

## Benchmarking of anti-TB drugs in a human airway air–liquid interface (ALI) system

To assess whether anti-TB compounds could inhibit extracellular Mtb replication or modulate epithelial cell responses to infection, we benchmarked their responses across the three drug categories. As such, Mtb-infected ALI cultures were exposed to anti-TB compounds, and bacterial load at the apical surface was measured 6 days later by both bioluminescence and CFU analyses. All standard-of-care drugs, except PZA, significantly reduced luminescence intensity, indicating that they effectively crossed the epithelial barrier to inhibit bacterial replication (Fig. 6A). In contrast, no effect was observed following treatment with host-directed therapies or virulence factor-targeting compounds. CFU analysis further supported these observations, revealing that only INH, RIF, and moxifloxacin (MXF) significantly suppressed bacterial growth (Fig. 6B). In parallel, LZD reduced luminescence but failed to lower viable counts (Fig. 6A and B). Together, these results demonstrate that the ALI model can effectively distinguish compounds with direct antibacterial activity from those that primarily target host responses or bacterial virulence mechanisms.

Finally, we investigated whether these compounds modulate the inflammatory response in ALI cultures, focusing on IL-6 and IL-8 secretion, which are linked to lung inflammation and TB pathology. As expected, Mtb infection led to a significant increase in IL-6 and IL-8 production by epithelial cells (Fig. 6C). Notably, despite their vigorous bactericidal activity, standard-of-care drugs did not reduce IL-6 secretion, whereas MXF and PZA modestly decreased IL-8 levels. Among all tested compounds, ETZ was unique in enhancing inflammation, reflected in a significant level of IL-6 production without influencing IL-8 (Fig. 6C). Although none of the host-directed therapies compounds significantly reduced the bacterial load, most of them modulated the cytokine profile in this system: IBU and DORA significantly reduced IL-6 secretion, while aspirin (ASA), IBU, and MET decreased IL-8 release (Fig. 6C). Collectively, these results demonstrate that the

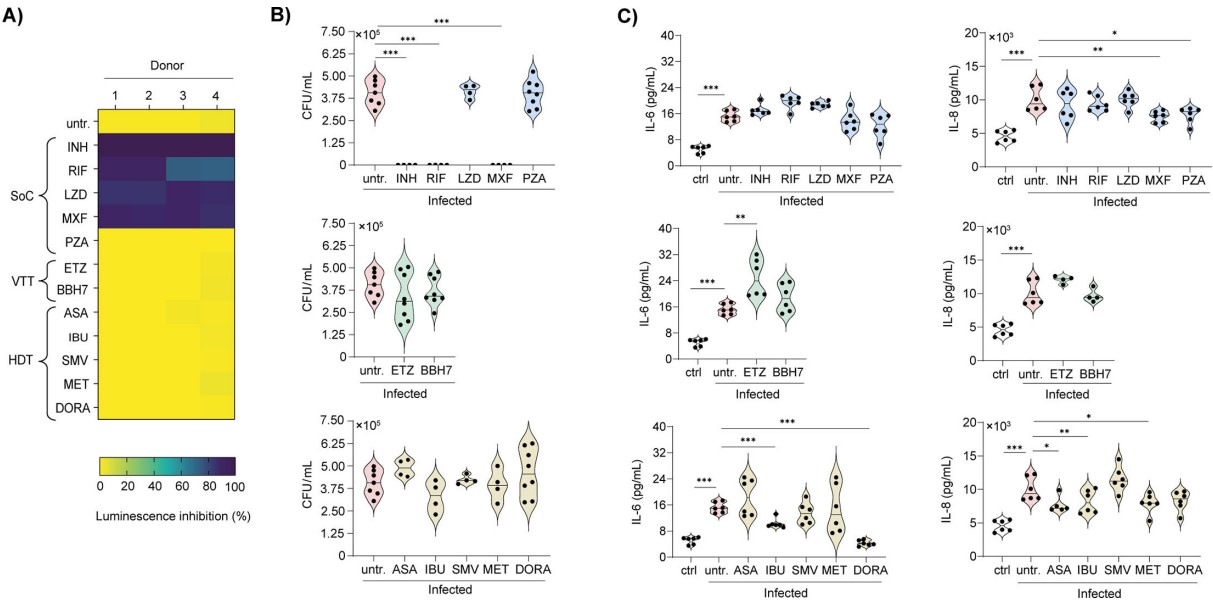

**FIG 6** Evaluation of bacterial control and host response modulation by anti-TB agents in human airway air–liquid interface (ALI) system. Airway ALI cultures were infected with H37Rv::lux (400,000 bac/insert) at the apical side, treated with different compounds (SoC, standard-of-care antibiotics; VTT, virulence-targeting therapies; HDT, host-directed therapies) in the basal compartment, and incubated for 6 days. (A) Heatmap showing luminescence inhibition. (B) Colony-forming unit (CFU) of Mtb-infected airway ALI model. (C) IL-6 and IL-8 production by the Mtb-infected airway ALI model. Compound concentrations tested were as follows: SoC (INH = 10 µM; RIF = 10 µM; LZD = 10 µM; MXF = 10 µM; PZA = 25 µM), VTT (ETZ = 10 µM; BBH7 = 10 µM), HDT (ASA = 50 µM; IBU = 50 µM; SMV = 10 µM; MET = 50 µM; DORA = 50 µM). Significant differences were determined using an unpaired, non-parametric Mann–Whitney test. $P < 0.05$ (*), $P < 0.01$ (**), $P < 0.001$ (***).

ALI model can capture drug-induced modulation of epithelial inflammatory responses independently of antibacterial efficacy.

In contrast to the AML system, where intracellular bacterial replication and macrophage heterogeneity shape drug responsiveness, the ALI model reveals how epithelial barriers and extracellular localization constrain Mtb survival while uncoupling antibacterial activity from inflammatory modulation. Together, these findings illustrate how the intracellular macrophage niche and the extracellular epithelial niche impose distinct constraints on Mtb behavior and drug susceptibility, underscoring the complementary value of the AML and ALI models

## DISCUSSION

In our study, we developed and tested a dual *in vitro* platform that includes human AML and airway ALI cultures. This setup is designed to evaluate various anti-TB compounds, including standard antibiotics, host-directed therapies, and strategies targeting virulent pathogenic factors. By mimicking different niches of the human lung that Mtb encounters during infection, particularly the airway spaces where this pathogen interacts with resident macrophages and epithelial cells, we can comprehensively study the effects of these compounds in an unprecedented manner.

The AML model closely mirrors the biology of primary hAMs. Adapted from previous reports (17, 18), this study recapitulates the key characteristics of primary hAMs, including strong expression of the transcription factor PPAR-γ and high levels of signature markers, including CD206, CD64, CD14, CD16, CD163, CD169, CXCR4, and PD-L1. These cellular findings are reinforced by an integrative transcriptomic comparison of AMLs and MDMs from the same differentiation framework (17, 18). AMLs primarily express tissue-resident and immunoregulatory modules, such as *MRC1* (CD206), *MARCO*, *FCGR3A* (CD16), and *CD274* (PD-L1), whereas MDMs tend to express pro-inflammatory and matrix-remodeling genes, including *TNF*, *FCAR*, *CD86*, and *MMP9*. Unbiased differential expression reveals that AML-upregulated genes, including *ABHD5* (which suppresses NF-κB-dependent MMPs), *AOC3* (involved in airway immune regulation), and *KLF4* (involved in STAT6-coupled activation), form a lung-adapted lipid-centric program (49, 50). In contrast, lower levels of *GALM* and *TRAF1* suggest distinct glucose metabolism and reduced NF-κB/inflammasome activity compared to MDMs. Gene ontology enrichment suggests that macrophages are shaped by surfactant and ongoing epithelial interactions in the alveolar niche, aligning with expectations of GPCR signaling, inflammatory response regulation, and lipid homeostasis. These characteristics align with transcriptomic and phenotypic analyses of hAM *in vivo*, in which PPAR-γ functions as the master regulator of lipid metabolism and anti-inflammatory programming (6).

Importantly, AML recapitulated the functional heterogeneity of AMs observed across species during Mtb infection. In the mouse model, CD38 stratified AMs into distinct subsets: CD38$^{high}$ cells exhibited enhanced microbicidal programs and reduced permissiveness, whereas CD38$^{low}$ AMs were comparatively permissive (51). In human MDMs differentiated with conditioned medium from Mtb-infected macrophages, we previously identified a CD16$^+$CD163$^+$MerTK$^+$ subset with an anti-inflammatory profile that was more permissive to Mtb infection (24). Strikingly, this population was also detected in the lungs of non-human primates, reinforcing its physiological relevance (24). Consistent with these findings, the AML model revealed a similar functional dichotomy defined by the partitioning of CD14 and CD16. Although CD38 did not distinguish resistant and permissive subsets as observed in mice, both CD16 subsets upregulated CD38 equally upon infection. The CD14$^+$CD16$^+$ subset displayed an M2-like cell-surface marker signature (e.g., CD163, MARCO, and CD209), along with higher baseline CD169 expression, which correlated with a higher burden of intracellular bacteria. In contrast, the CD14$^+$CD16$^-$ subset showed an M1-like profile (e.g., CD86) and reduced susceptibility to infection. Importantly, these AML subsets are best interpreted as dynamic activation states induced by infection rather than as stable or ontogenetically distinct macrophage populations. Moreover, by high-dimensional flow cytometry

analysis, we revealed distinct activation states in macrophages driven by infection, underscoring the AML model's ability to recapitulate the complex phenotypic diversity observed in macrophages across lung niches during Mtb infection *in vivo* (7, 52). To further contextualize our results within a physiological framework, we integrated data from freshly isolated hAMs infected *ex vivo* with Mtb (38). Infected hAMs upregulated activation markers (e.g., *CD274* [PD-L1], *TNF*, *IL6*) while progressively losing anti-inflammatory regulator *FCGR3A* (CD16). This shift was accompanied by the induction of inflammatory chemokines, such as *CCL20*, *CCL4*, and *CXCL10*, which coordinate immune cell recruitment but can also drive tissue pathology. In addition, a strong induction of ISGs (e.g., *IFIT1*, *ISG15*, *MX1*, *OAS3*, *RSAD2*) underscored a dominant IFN-I axis, while regulators, such as *IRF7* and *NEURL3*, suggest layered control of this program. Thus, our data establish AMLs as a model that not only mirrors the functional dichotomy found in AMs in the lung but also reveals the layered activation states that govern host–pathogen interactions during Mtb infection. This complexity provides a physiologically relevant platform, particularly valuable for identifying new compounds that modulate host–pathogen interactions in TB.

To explore this potential, we benchmarked various anti-TB drugs using the AML model. We observed significant differences in their antibacterial and immunomodulatory properties. While most standard-of-care antibiotics effectively reduced intracellular bacterial load in AMLs, PZA's activity was initially underestimated by luminescence and later confirmed by CFU, highlighting the need for multiple, complementary assays. Importantly, the Lux signal (RLU) is a metabolic proxy and does not directly quantify colony-forming capacity; instead, it reflects luminescence-based reporter activity, which indirectly reflects bacterial metabolic and translational activity. These data show that PZA can impair bacterial viability without immediately suppressing detectable metabolic activity by the Lux system, underscoring the complementary nature and limitations of luminescence-based readouts relative to CFU enumeration. Interestingly, Heifets and colleagues reported that PZA exerts neither bacteriostatic nor bactericidal effects against Mtb in MDMs, regardless of macrophage activation status (53). This finding aligns with other pioneering studies that have established PZA as primarily effective *in vivo* but poorly potent *in vitro* (54–57). Indeed, multiple studies have demonstrated that PZA activity critically depends on acidic microenvironments that promote weak-acid permeation and disruption of intrabacterial pH homeostasis, and that intracellular localization within acidified phagosomes enhances PZA accumulation and efficacy. These observations suggest that variations in macrophage differentiation state, phagosomal maturation, and subcellular niche composition may profoundly influence PZA responsiveness *in vitro* (58–60), thereby providing a mechanistic framework to interpret the differential activity observed in AMLs. Yet, the significant effect of PZA we now report in AMLs is probably reflective of their hAM biology, which is known for its weaker microbicidal activity in early Mtb containment (6, 7, 51, 52, 61). While direct comparisons of PZA efficacy in AML versus MDM models (derived from the same donor) remain to be explored, the hAM-like phenotype of AMLs suggests that they better recapitulate lung-specific drug responses targeting the alveolar compartment.

In addition, host-directed therapies and treatments targeting virulence factors in infected AMLs showed variable antibacterial effects, often modulating inflammatory signals, such as TNF-α and IL-6. Notably, several of these interventions reduced bacterial burden as measured by CFU without eliciting marked changes in luminescence-based readouts. This discrepancy likely reflects the indirect mechanisms of action of host-directed therapies, which enhance macrophage antimicrobial functions, such as phagolysosomal maturation, autophagy, or metabolic reprogramming, without directly impairing bacterial metabolic activity. As a result, bacterial survival is restricted, while translation-dependent metabolic reporters remain largely unaffected. Studies using MDMs or THP-1 cells similarly report host-directed therapies targeting inflammatory pathways (e.g., autophagy inducers, metabolic shifting agents), but often require exaggerated doses to modulate the cytokine response (62). Moreover, the reported impact of

virulence-targeting factors on immune signaling appears to be highly target-specific, with some factors attenuating IFN-I production (via cGAS) by blocking the secretion of EsxA/EsxB. In contrast, others modulate virulence-associated transcriptional programs without apparent suppression of cytokine release (63). Yet, systematic evaluations of immune modulation mediated by virulence-targeting factors in human lung-resident macrophages remain scarce. Taken together, the observed uncoupling between bacterial control and cytokine modulation in AMLs indicates that this cell model is sensitive to host-directed therapies and interventions targeting virulence factors and underscores its value as a physiologically relevant platform for targeting host–pathogen interactions aligned with hAM biology. Notably, recent findings validating iPSC-derived macrophages as a platform for Mtb infection and drug testing (64) have highlighted their scalability and physiological relevance for high-throughput screening (HTS). Our data further emphasize that AMLs exhibit a phenotype more closely resembling *in vivo* alveolar macrophages and offer additional practical advantages, including faster and easier generation, the ability to capture interindividual variability in treatment responses, and potential for HTS applications.

Another key aspect of this study is our focus on the airway epithelium, which is increasingly recognized as vital in modulating immune responses during active TB. In line with recent reports (22, 47), we found that although Mtb can colonize the apical surface of airway epithelial cells, it fails to traverse the epithelial barrier or replicate robustly, highlighting the airway epithelium's intrinsic defensive capacity. This observation aligns with clinical and experimental evidence suggesting that airway epithelial cells act as gatekeepers through mucociliary clearance and innate defense pathways, thereby limiting bacterial dissemination into deeper tissues (47, 65, 66). Interestingly, bacterial loads in the ALI system naturally decline over 6 days of infection, likely due to epithelial innate immune mechanisms, including the production of antimicrobial peptides such as β-defensin-1 and RNase 7, as we previously reported in Mtb-infected airway organoids (26). Our results demonstrate a clear discriminating capacity between treatments: first-line drugs such as INH, RIF, and MXF significantly suppressed bacterial growth, whereas LZD, a bacteriostatic antibiotic, efficiently impairs protein synthesis, as shown by a substantial reduction of luminescence but a failure to impact CFUs, reflecting discrepancies between metabolic readouts and bacterial viability assays, as also noted in macrophage-based assays. This effect is mechanistically consistent with linezolid's known mode of action, which inhibits bacterial protein synthesis by targeting the 50S ribosomal subunit. Because luminescence depends on active protein translation, ribosomal inhibition is expected to strongly reduce the luminescent signal without necessarily affecting bacterial viability, as evidenced by unchanged CFU counts. This apparent lack of effect likely reflects the intrinsic limitation of bacterial growth in the ALI system, where bacterial loads naturally decline over time, thereby masking any additional contributions from bacteriostatic drug activity. PZA was ineffective, consistent with previous studies reporting its limited or delayed activity *in vitro* due to pH-dependent activation. This underscores the translational potential of the ALI system in predicting which drugs can access and remain active at the airway interface.

The ALI model also facilitated the assessment of epithelial-driven inflammatory responses and their modulation through therapeutic interventions. As expected, Mtb infection led to significant secretion of pro-inflammatory cytokines IL-6 and IL-8, aligning with their established roles in driving inflammation and leukocyte recruitment in TB pathology. Although standard-of-care antibiotics exhibited the predicted bactericidal properties, they did not significantly reduce cytokine secretion, indicating that merely clearing the pathogen is insufficient to mitigate epithelial-driven inflammation. This pattern is consistent with features of the airway ALI context, where Mtb largely remains at the apical surface and epithelial signaling is dominated by apical and endosomal PRRs (e.g., TLRs and C-type lectins) (22, 47). In contrast, macrophages additionally integrate cytosolic cues delivered by virulent Mtb via ESX-1 secretion system, which exports virulence factors such as ESAT-6 that permeabilize the phagosomal membrane, allowing

Mtb antigens to access the host cytosol and activate cytosolic sensors (e.g., cGAS/STING, AIM2, NLRs, RIG) (66–70). Accordingly, most antibiotic therapies kill Mtb but leave behind PAMP-rich debris (lipoproteins, cell wall lipids, nucleic acids) that continues to engage surface and endosomal PRRs (71, 72). By contrast, robust activation of cytosolic sensors typically requires live, ESX-1-competent bacilli, an event unlikely to occur once bacteria are non-viable under antibiotic treatment (67, 73). Together, these observations suggest the need for adjunct host-directed strategies that modulate epithelial sensing pathways in addition to pathogen killing.

Conversely, ETZ uniquely increased IL-6 secretion in the ALI system, suggesting possible off-target pro-inflammatory effects. By comparison, host-directed therapies, such as IBU, ASA, MET, and DORA, effectively attenuated IL-6 and/or IL-8 levels without altering bacterial load, demonstrating the ALI model's ability to capture immunomodulatory effects independent of antimicrobial activity. Collectively, these findings reinforce the ALI model's relevance as a platform for exploring the antimicrobial efficacy and immunomodulatory potential of anti-TB compounds in the airway context. By emphasizing the extracellular niche of Mtb during active TB, rather than the purely intracellular dynamics captured by macrophage-based models, the ALI system highlights the airway epithelium's dual role as a barrier and as an active regulator of inflammation. Importantly, excessive inflammation is a hallmark of severe TB and is strongly associated with tissue destruction and disease progression (74). Thus, therapeutic strategies that combine pathogen clearance with modulation of host inflammatory pathways may be particularly valuable, especially in advanced or disseminated TB, where hyperinflammation exacerbates pathology and worsens clinical outcomes. Our complementary macrophage and ALI models provide a unique framework to dissect these processes. While AMLs capture intracellular niches where antibiotics and host-directed agents can reduce both bacterial burden and pro-inflammatory signaling, ALIs reflect epithelial-driven responses that remain largely unaffected by bacterial killing alone. Together, these models demonstrate that compounds can differentially affect intracellular versus extracellular bacteria and modulate immune responses in cell types that sense and respond to Mtb in fundamentally distinct ways. This underscores the therapeutic potential of combining antimicrobial activity with host-directed modulation to more effectively treat severe TB. These key disease parameters may constitute valuable predictive indicators in human settings for compound down-selection, and further confirmation of their potential benefits in *in vivo* models of severe TB is warranted. While pro-inflammatory cytokines contribute to tissue damage when produced in excess, they are also essential for bacterial control, and their disproportionate suppression may favor mycobacterial persistence or outgrowth. Therefore, achieving a balanced modulation of host inflammatory responses is likely critical for therapeutic benefit.

In summary, our study validates and highlights the relevance of using a dual *in vitro* system that integrates AML and airway ALI responses as a biologically relevant framework for dissecting the intracellular and extracellular niches of Mtb within the lung. By recapitulating key features of primary human airway mucosa (hAMs) and the epithelial barrier, this approach enables a side-by-side assessment of the antimicrobial potency and immunomodulatory capacity of anti-TB compounds. The AML model revealed that macrophage activation states, particularly those associated with CD16 expression, critically influence bacterial burden and immune activation. In contrast, the ALI system highlighted the impact on epithelial-driven inflammation. Nevertheless, we emphasize that this platform remains an *in vitro* model that mimics the human lung microenvironment; therefore, it should be viewed not as a replacement, but as a complementary approach that may help link responses observed in preclinical models to those in humans. In addition, we stress that *in vivo* studies remain essential to determine whether these immunomodulatory profiles are preserved within a physiological context that allows interactions with adaptive immune responses, as well as other systemic and tissue-level regulatory components, and whether they ultimately translate into therapeutic benefit. Collectively, these findings position AML and ALI cultures as

complementary, tractable models that may bridge the gap between reductionist *in vitro* assays and the complex human *in vivo* lung environment, offering a valuable system for advancing both drug discovery and host-directed therapeutic strategies in TB.

## ACKNOWLEDGMENTS

We would like to acknowledge Myriam Ben-Neji and Genotoul Anexplo-IPBS for access to the BSL-3 facilities; the Genotoul TRI-IPBS facilities for imaging; and the histology platform at IPBS. We thank Vanessa Soldan and Stephanie Balor (CBI) for TEM, Isabelle Fourquaux (CMEAB) for SEM work, and Yoann Rombouts and Claude Gutierrez for obtaining the integrative plasmid pMV30-hsp-Lux13 (Addgene #26161) and constructing the strain H37Rv::lux, respectively. We thank Jean-Philippe Girard for giving access to the Zeiss Axio imager M2 microscope (IPBS). We thank Drs. Aurelien Brindel, Gavin Plat, and Valentin Heluain from Toulouse Hospital for BAL patient samples. Additionally, we thank all members of the ITHEMYC Consortium (https://tbvi.eu/our-projects/ithemyc/) for their scientific input, and in particular the Global Health Medicines R&D and Sample Management departments at GLAXOSMITHKLINE INVESTIGACION Y DESARROLLO SL (GSK) for providing the benchmarking compounds.

The ITHEMYC project, "Novel immunotherapies for tuberculosis and other mycobacterial diseases," is funded by the European Union's Health and Digital Executive Agency (HaDEA) under Grant Agreement number 101080462 and by UK Research and Innovation (UKRI). This work was also supported by the Centre National de la Recherche Scientifique, the Université Paul Sabatier, the Institut National de la Santé et de la Recherche Médicale, the Agence Nationale de la Recherche, the Agence Nationale de Recherche sur le Sida et les hépatites virales (ANRS MIE), Sidaction, and the Fondation pour la Recherche Médicale. Postdoctoral fellows C.C.B.B., T.B., and J.M.S-L. are recipients of the ITHEMYC project (Grant Agreement number 101080462), the Bill and Melinda Gates Foundation INV-046428, and SECTEI/071/2024 (Mexican postdoctoral scholarship in academic or research institutions abroad with international recognition) scholarships, respectively. A doctoral fellowship from ANRS MIE supported N.F.

Conceptualization: C.C.B.B., N.F., F.D., C.V., G.L.-V., and C.C. Methodology: C.C.B.B., N.F., T.B., J.M.S.-L., E.N., P.V., A.M., R.P., C.V., G.L.-V., and C.C. Investigation: C.C.B.B., N.F., T.B., M.P., B.S., D.P., J.M.S.-L., B.M.A.-C., E.N., P.V., A.M., and C.C. Resources: E.N., N.G., J.M., R.P., B.R.-M., F.D., O.N., E.M., C.V., G.L.-V., R.V., and C.C. Formal Analysis: C.C.B.B., N.F., T.B., M.P., J.M.S.-L., B.M.A.-C., E.N., P.V., G.L.-V., and C.C. Writing: C.C.B.B., N.F., G.L.V., and C.C. Visualization: C.C.B.B., N.F., T.B., M.P., J.M.S.-L., B.M.A.-C., E.N., P.V., A.M., R.P., C.V., and C.C. Supervision: F.D., C.V., E.M., G.L.-V., and C.C. Project Administration: C.C. Funding acquisition: R.P., O.N., E.M., C.V., G.L.-V., and C.C. Corresponding authors: C.C.B.B. and C.C. are responsible for the ownership and responsibility that are inherent to all aspects of this study.

## AUTHOR AFFILIATIONS

[1]Institut de Pharmacologie et de Biologie Structurale (IPBS), Université de Toulouse, CNRS, Toulouse, France
[2]International Research Project CNRS "MAC-TB/HIV", Toulouse, France
[3]Unidad de Investigación Médica en Genética Humana (UIMGH), Hospital de Pediatría Centro Médico Nacional Siglo XXI, Instituto Mexicano del Seguro Social (IMSS), Mexico City, México
[4]Children's Hospital of Tlaxcala, Tlaxcala, México
[5]Service de Pneumologie, Hôpital Larrey, CHU de Toulouse, Toulouse, France
[6]Service de Chirurgie Thoracique, Hôpital Larrey, CHU de Toulouse, Toulouse, France

## AUTHOR ORCIDs

Caio César Barbosa Bomfim http://orcid.org/0000-0003-1998-9237
Natacha Faivre http://orcid.org/0009-0008-0330-5519
Thomas Benoist http://orcid.org/0000-0003-0116-2629

José Manuel Sánchez-López  http://orcid.org/0000-0002-1349-4746
Beatriz Melissa Aponte-Castillo  http://orcid.org/0009-0004-5613-4718
Pénélope Viana  http://orcid.org/0000-0003-0098-4831
Arnaud Métais  http://orcid.org/0000-0002-9538-0338
Renaud Poincloux  http://orcid.org/0000-0003-2884-1744
Brigitte Raynaud-Messina  http://orcid.org/0009-0000-1512-0871
Fabrice Dumas  http://orcid.org/0000-0002-9164-4527
Olivier Neyrolles  http://orcid.org/0000-0003-0047-5885
Christel Vérollet  http://orcid.org/0000-0002-1079-9085
Etienne Meunier  http://orcid.org/0000-0002-3651-4877
Geanncarlo Lugo-Villarino  http://orcid.org/0000-0003-4620-8491
Céline Cougoule  http://orcid.org/0000-0002-6795-5448

## FUNDING

| Funder | Grant(s) | Author(s) |
| --- | --- | --- |
| Bill and Melinda Gates Foundation | INV-046428 | Céline Cougoule |
| Sidaction | 22-2-AEQ-13457-1 | Christel Vérollet |
| H2020 Health | 101080462 | Céline Cougoule |
| UK Research and Innovation | 101080462 | Céline Cougoule |
| Fondation pour la Recherche Médicale | EQU202303016313 | Christel Vérollet |
| Agence Nationale de Recherches sur le Sida et les Hépatites Virales | 2022-ECTZ190463, 2022-ECTZ205302, 2023-ECTZ293306 | Christel Vérollet |

## AUTHOR CONTRIBUTIONS

José Manuel Sánchez-López, Formal analysis | Beatriz Melissa Aponte-Castillo, Formal analysis | Nicolas Guibert, Resources | Julien Mazières, Resources.

## ADDITIONAL FILES

The following material is available online.

### Supplemental Material

**Fig. S1 (Spectrum03729-25-s0001.tif).** Molecular characterization of AML and MDM and flow cytometry gating strategy for phenotypic analysis.
**Fig. S2 (Spectrum03729-25-s0002.tif).** Differential gene expression analysis of AML versus MDM.
**Fig. S3 (Spectrum03729-25-s0003.tif).** Optimization of infection conditions and phenotypic profiling of AML cells upon Mtb infection.
**Fig. S4 (Spectrum03729-25-s0004.tif).** Evaluation of toxicity induced by anti-TB agents in AML cells.
**Supplemental material (Spectrum03729-25-s0005.docx).** Supplemental figure legends; Tables S1 to S4.
**Movie S1 (Spectrum03729-25-s0006.mp4).** Ciliary beating of airway ALI culture visualized using light microscopy with a 60× objective.

### Open Peer Review

**PEER REVIEW HISTORY (review-history.pdf).** An accounting of the reviewer comments and feedback.

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
