## [Reviewer comments · Microbiology Spectrum]

Microbiology Spectrum

Dual human lung models reveal compartment-specific activity of anti-tuberculosis drugs and host-directed therapies

Caio Barbosa Bomfim, Natacha Faivre, Thomas Benoist, Manon Popis, Bastien Suire, David Pericat, José Manuel Sánchez-López, Beatriz Melissa Aponte-Castillo, Emmanuelle Näser, Pénélope Viana, Nicolas Guibert, Romain Vergé, Julien Mazières, Arnaud Métais, Renaud Poincloux, Brigitte Raynaud-Messina, Fabrice Dumas, Olivier Neyrolles, Christel Vérollet, Etienne Meunier, Geanncarlo Lugo-Villarino, and Celine Cougoule

Corresponding Author(s): Caio Barbosa Bomfim, Institut de pharmacologie et biologie structurale

Review Timeline:

Submission Date:	November 18, 2025
Editorial Decision:	January 5, 2026
Revision Received:	March 2, 2026
Accepted:	March 18, 2026

Editor: Sophia Georghiou

Reviewer(s): Disclosure of reviewer identity is with reference to reviewer comments included in decision letter(s). The following individuals involved in review of your submission have agreed to reveal their identity: Susanta Pahari (Reviewer #2)

Transaction Report:

DOI: <https://doi.org/10.1128/spectrum.03729-25>

Re: Spectrum03729-25 (Dual human lung models reveal compartment-specific activity of anti-tuberculosis drugs and host-directed therapies)

Dear Dr. Caio Cesar Barbosa Bomfim:

Thank you for the privilege of reviewing your work. Below you will find my comments, instructions from the Spectrum editorial office, and the reviewer comments.

Revision Guidelines

Sincerely,
Sophia Georghiou
Editor
Microbiology Spectrum

Reviewer #1 (Comments for the Author):

In this manuscript from Caio Cesar Barbosa Bomfim and colleagues, the authors propose that human lung models integrating alveolar macrophage-like (AML) cells and airway air-liquid interface (ALI) could represent promising alternative model for drug discovery in TB research. They first set up a system where they differentiate CD14+ monocytes into AML cells, and perform several phenotypic assays to demonstrate that this system could be used as a promising laboratory system that mimics primary alveolar macrophages including the CD16+, which are clinically relevant. Then, they use the ALI system to provide an alternative biological model that might benefit drug discovery studies and try to provide a proof of concept that this model could benefit the

scientific community.

Altogether, this study is very timely as more reliable models are really needed to better understand Mtb pathogenesis and further identify drug combinations that are highly effective against the tubercle bacilli. However, the study fails to really convince, with numerous discrepancies and inconsistencies. More control experiments would benefit the story and help to better support the authors' conclusions.

Here are some of the points that need to be carefully addressed by the authors:

Major Points

Fig2. The TEM Micrographs from AML CTRL and Mtb-infected conditions are quite puzzling, as they appear from the reviewer's understanding already infected by another bacterial species that displays a more coccoidal shape and seems to replicate efficiently within AML, as they show some clearly visible division septa. I was unsure why MDM differentiation was carried out without Pen-Strep, while it was used for AML. It seems that AML differentiation process might have affect the sterility of the culture? and that some of the samples might have been contaminated. Can the authors comment on this?

If I'm wrong, please try to identify what are these structures with their biological relevance, but if this is the case, I urge the authors to be as transparent as possible and discuss this in their manuscript as it seemed voluntarily unmentioned in the current version. In light of this, it is essential that the authors provide clear evidence that only the TEM samples were affected by this contamination? And more importantly critically think about how this potential contamination can impact all the subsequent results obtained in their manuscript with AML, from cell viability to transcriptomic, and anti-TB susceptibility profiling. Can the authors thoroughly comment on this? As this is questioning a large part of the manuscript, including its biological relevance and the conclusions drew by the authors.

Fig4. Lines 508-509 - The authors report that as expect anti-TB drug are effective except PZA that does not affect replication dynamics as shown in their Heat-Map with an approximate 0% inhibition across 7 donors. Then, Lines 513-514 they state the following "All standard-of-care antibiotics, including PZA, significantly suppressed bacterial replication." So PZA does not affect replication dynamics with the Lux Reporter, but affects viability according to CFU counting assay with a 50% reduction in CFU. Can the authors comment on these discrepancies? And discuss about how reliable the Lux system is? Have they performed a calibration standard to correlate luminescence intensity with CFU? The timepoint T0 that corresponds to the initial inoculum in CFU or Luminescence Units should be also available for the readers, to assess the replication dynamics over 6 days as done in Fig5H.

Also, its absolutely not clear what concentration of drugs were used in this assay. Please provide more information about how these experiments were performed.

Of note, a thorough side-by-side comparison of Mtb replication dynamics overtime and antibiotic efficacy in hAM vs AML vs ALI would definitely increase the strength of this study, and showcase how AML and ALI can be complementary and suitable model for TB-Host interactions and drug discovery studies.

Fig5 Lines 569-572 the authors state that "Strikingly, rather than replicating, Mtb showed a progressive decline over time in both luminescence and viable counts, indicating not only a lack of replication but a potential loss of bacterial viability in this model". Have the authors performed any statistics to confirm that the decrease they observed is significant? As it seems that this decrease remains minimal with close to half a Log10.

Fig6. All the concerns about Fig4 applies to this Figure as well (T0 CFU, Concentration of antibiotics used etc). It is essential to define whether antibiotics are simply blocking replication or have a bactericidal effect. In addition, can the authors comments about the discrepancies they have in this system with LZD which is showing 70-100% inhibition using the Mtb Lux system but no changes in term of CFU.

Minor Point:

Have the authors considered using an aminoside in their assays with AML and ALI as control as they are known for their poor permeability within cells, but good efficacy against free extracellular bacteria?

Reviewer #2 (Public repository details (Required)):

NCBI GEO (Gene Expression Omnibus) platform using the accession numbers GSE188945 and GSE189996

Reviewer #2 (Comments for the Author):

This manuscript presents a well-designed and biologically relevant dual human lung platform that integrates alveolar macrophage-like (AML) cells with airway air-liquid interface (ALI) cultures to evaluate anti-tuberculosis drugs and host-directed therapies. The models are carefully characterized, and the compartment-specific comparison of intracellular macrophage responses and airway epithelial responses represents a meaningful advance over conventional in vitro systems. The study is technically strong and addresses an important gap in TB drug evaluation. Importantly, the authors recapitulate previously published AML models developed by other groups, providing appropriate validation and increasing confidence in the robustness and reproducibility of the platform. That said, several points should be addressed to further strengthen the manuscript and clarify its impact.

1. The Introduction is thorough and well-referenced but would benefit from greater concision. The central hypothesis and specific objectives should be stated more explicitly near the end of the section. In addition, briefly clarifying how the AML and ALI models are intended to complement each other would help orient readers before the Results.

2. The AML model is convincingly validated, and the identification of infection-associated heterogeneity within AMLs is a notable strength. However, several Results sections are overly detailed, particularly those describing transcriptomic analyses, and could be streamlined. The interpretation of discrepancies between luminescence-based readouts and CFU measurements should be clarified. It would also be helpful to more clearly state whether the identified AML subsets represent stable populations or dynamic infection-induced states.
3. The Discussion effectively integrates the findings and places them in the context of current TB literature.
4. That said, conclusions about host-directed therapies should be made more cautiously, since immunomodulatory effects do not always indicate therapeutic benefit. The model limitations, including the lack of adaptive immune components, could be more openly acknowledged.
5. While the results are clearly presented, the section could be strengthened by placing more emphasis on conceptual synthesis to integrate the findings, rather than restating individual observations.
6. A significant limitation is that most conclusions remain superficial. The study catalogs differences in bacterial load and cytokine production across models but offers little mechanistic explanation for these differences. For example, pyrazinamide activity in AMLs but not in ALI cultures is interpreted as niche-specific efficacy, but no direct evidence is provided regarding intracellular pH, drug activation, or bacterial metabolic state in AMLs versus epithelial surfaces. Further explanation would help.
7. CD14/CD16-based stratification is informative, but the relationship to bona fide resident versus recruited AM populations in vivo remains speculative.
8. The manuscript acknowledges this issue but still draws conclusions about bacteriostatic versus bactericidal activity, particularly for linezolid.
9. Cytokine measurements (TNF- α , IL-6, IL-8) are used as proxies for disease severity, but the biological implications are not always clear. Reduced cytokine production is implicitly framed as beneficial, yet excessive suppression of TNF or IL-6 could also impair bacterial control in vivo.

Minor Comments

- The statistical methods should clarify how multiple comparisons were handled across the large drug panels.
- *Mycobacterium tuberculosis* should be italicized throughout the manuscript.
- An asterisk (*) is missing in the Figure 1 legend.
- Reporting bacterial burden as CFU per well may be misleading; for in vitro assays, CFU is conventionally expressed as CFU/mL.
- Cytokine concentrations should be reported using a consistent unit (either pg/mL or ng/mL) across all figures and text.

In this manuscript from Caio Cesar Barbosa Bomfim and colleagues, the authors propose that human lung models integrating alveolar macrophage-like (AML) cells and airway air-liquid interface (ALI) could represent promising alternative model for drug discovery in TB research. They first set up a system where they differentiate CD14+ monocytes into AML cells, and perform several phenotypic assays to demonstrate that this system could be used as a promising laboratory system that mimics primary alveolar macrophages including the CD16+, which are clinically relevant. Then, they use the ALI system to provide an alternative biological model that might benefit drug discovery studies and try to provide a proof of concept that this model could benefit the scientific community.

Altogether, this study is very timely as more reliable models are really needed to better understand Mtb pathogenesis and further identify drug combinations that are highly effective against the tubercle bacilli. However, the study fails to really convince, with numerous discrepancies and inconsistencies. More control experiments would benefit the story and help to better support the authors' conclusions.

Here are some of the points that need to be carefully addressed by the authors:

Major Points

Fig2. The TEM Micrographs from AML CTRL and Mtb-infected conditions are quite puzzling, as they appear from the reviewer's understanding already infected by another bacterial species that displays a more coccoidal shape and seems to replicate efficiently within AML, as they show some clearly visible division septa. I was unsure why MDM differentiation was carried out without Pen-Strep, while it was used for AML. It seems that AML differentiation process might have affect the sterility of the culture? and that some of the samples might have been contaminated. Can the authors comment on this?

If I'm wrong, please try to identify what are these structures with their biological relevance, but if this is the case, I urge the authors to be as transparent as possible and discuss this in their manuscript as it seemed voluntarily unmentioned in the current version. In light of this, it is essential that the authors provide clear evidence that only the TEM samples were affected by this contamination? And more importantly critically think about how this potential contamination can impact all the subsequent results obtained in their manuscript with AML, from cell viability to transcriptomic, and anti-TB susceptibility profiling. Can the authors thoroughly comment on this? As this is questioning a large part of the manuscript, including its biological relevance and the conclusions drew by the authors.

Fig4. Lines 508-509 - The authors report that as expect anti-TB drug are effective except PZA that does not affect replication dynamics as shown in their Heat-Map with an approximate 0% inhibition across 7 donors. Then, Lines 513-514 they state the following "All standard-of-care antibiotics, including PZA, significantly suppressed bacterial replication." So PZA does not affect replication dynamics with the Lux Reporter, but affects viability according to CFU counting assay with a 50% reduction in CFU. Can the authors comment on these discrepancies? And discuss about how reliable the Lux system is? Have they performed a calibration standard to correlate luminescence intensity with CFU? The timepoint T0 that corresponds to the initial inoculum in CFU or Luminescence Units should be also available for the readers, to assess the replication dynamics over 6 days as done in Fig5H.

Also, its absolutely not clear what concentration of drugs were used in this assay. Please provide more information about how these experiments were performed.

Of note, a thorough side-by-side comparison of Mtb replication dynamics overtime and antibiotic efficacy in hAM vs AML vs ALI would definitely increase the strength of this study, and showcase how AML and ALI can be complementary and suitable model for TB-Host interactions and drug discovery studies.

Fig5 Lines 569-572 the authors state that “Strikingly, rather than replicating, Mtb showed a progressive decline over time in both luminescence and viable counts, indicating not only a lack of replication but a potential loss of bacterial viability in this model”. Have the authors performed any statistics to confirm that the decrease they observed is significant? As it seems that this decrease remains minimal with close to half a Log10.

Fig6. All the concerns about Fig4 applies to this Figure as well (T0 CFU, Concentration of antibiotics used etc). It is essential to define whether antibiotics are simply blocking replication or have a bactericidal effect. In addition, can the authors comments about the discrepancies they have in this system with LZD which is showing 70-100% inhibition using the Mtb Lux system but no changes in term of CFU.

Minor Point:

Have the authors considered using an aminoglycoside in their assays with AML and ALI as control as they are known for their poor permeability within cells, but good efficacy against free extracellular bacteria?

Reviewer Report

This manuscript presents a well-designed and biologically relevant dual human lung platform that integrates alveolar macrophage-like (AML) cells with airway air-liquid interface (ALI) cultures to evaluate anti-tuberculosis drugs and host-directed therapies. The models are carefully characterized, and the compartment-specific comparison of intracellular macrophage responses and airway epithelial responses represents a meaningful advance over conventional in vitro systems. The study is technically strong and addresses an important gap in TB drug evaluation. Importantly, the authors recapitulate previously published AML models developed by other groups, providing appropriate validation and increasing confidence in the robustness and reproducibility of the platform. That said, several points should be addressed to further strengthen the manuscript and clarify its impact.

1. The Introduction is thorough and well-referenced but would benefit from greater concision. The central hypothesis and specific objectives should be stated more explicitly near the end of the section. In addition, briefly clarifying how the AML and ALI models are intended to complement each other would help orient readers before the Results.
2. The AML model is convincingly validated, and the identification of infection-associated heterogeneity within AMLs is a notable strength. However, several Results sections are overly detailed, particularly those describing transcriptomic analyses, and could be streamlined. The interpretation of discrepancies between luminescence-based readouts and CFU measurements should be clarified. It would also be helpful to more clearly state whether the identified AML subsets represent stable populations or dynamic infection-induced states.
3. The Discussion effectively integrates the findings and places them in the context of current TB literature.
4. That said, conclusions about host-directed therapies should be made more cautiously, since immunomodulatory effects do not always indicate therapeutic benefit. The model limitations, including the lack of adaptive immune components, could be more openly acknowledged.
5. While the results are clearly presented, the section could be strengthened by placing more emphasis on conceptual synthesis to integrate the findings, rather than restating individual observations.
6. A significant limitation is that most conclusions remain superficial. The study catalogs differences in bacterial load and cytokine production across models but offers little mechanistic explanation for these differences. For example, pyrazinamide activity in AMLs but not in ALI cultures is interpreted as niche-specific efficacy, but no direct evidence is provided regarding intracellular pH, drug activation, or bacterial metabolic state in AMLs versus epithelial surfaces. Further explanation would help.
7. CD14/CD16-based stratification is informative, but the relationship to bona fide resident versus recruited AM populations in vivo remains speculative.
8. The manuscript acknowledges this issue but still draws conclusions about bacteriostatic versus bactericidal activity, particularly for linezolid.
9. Cytokine measurements (TNF- α , IL-6, IL-8) are used as proxies for disease severity, but the biological implications are not always clear. Reduced cytokine production is implicitly framed as beneficial, yet excessive suppression of TNF or IL-6 could also impair bacterial control in vivo.

Minor Comments

- The statistical methods should clarify how multiple comparisons were handled across the large drug panels.
- *Mycobacterium tuberculosis* should be italicized throughout the manuscript.
- An asterisk (*) is missing in the Figure 1 legend.
- Reporting bacterial burden as CFU per well may be misleading; for in vitro assays, CFU is conventionally expressed as CFU/mL.
- Cytokine concentrations should be reported using a consistent unit (either pg/mL or ng/mL) across all figures and text.

RESPONSE TO REVIEWERS

REVIEWER 1

Major Points

1) Fig. 2. The TEM Micrographs from AML CTRL and Mtb-infected conditions are quite puzzling, as they appear from the reviewer's understanding already infected by another bacterial species that displays a more coccoidal shape and seems to replicate efficiently within AML, as they show some clearly visible division septa. I was unsure why MDM differentiation was carried out without Pen-Strep, while it was used for AML. It seems that AML differentiation process might have affect the sterility of the culture? and that some of the samples might have been contaminated. Can the authors comment on this? If I'm wrong, please try to identify what are these structures with their biological relevance, but if this is the case, I urge the authors to be as transparent as possible and discuss this in their manuscript as it seemed voluntarily unmentioned in the current version. In light of this, it is essential that the authors provide clear evidence that only the TEM samples were affected by this contamination? And more importantly critically think about how this potential contamination can impact all the subsequent results obtained in their manuscript with AML, from cell viability to transcriptomic, and anti-TB susceptibility profiling. Can the authors thoroughly comment on this? As this is questioning a large part of the manuscript, including its biological relevance and the conclusions drew by the authors.

We sincerely thank the reviewer for this careful and critical assessment of Fig. 2. We would also like to apologize for not having immediately recognized that the structures observed in the original TEM images could potentially indicate contamination in the experiment. We are grateful that the reviewer pointed this out. We fully agree that it requires clarification.

To address this major concern, we carefully re-evaluated the original samples and confirmed that the 24h infection experiment with AMLs and Mtb was indeed contaminated. We thus repeated the entire TEM experiments on two independent donors under strictly controlled conditions. We systematically performed CFU analyses, before TEM sample fixation, to verify the sterility and specificity of the Mtb infection. Importantly, in these repeated experiments, the only colony-forming units detected on agar plates incubated at 37°C for 3 weeks corresponded to *Mycobacterium tuberculosis* (Mtb), with no evidence of any other fast-growing bacterial or yeast species.

In the revised manuscript, we are submitting new TEM images generated from these repeated and controlled experiments, in which we no longer detect the previously observed coccoidal structures or septum-like features that had raised the reviewer's concern (Fig. 2A). The absence of these structures in the repeated experiments supports the interpretation of the reviewer that they were not intrinsic to alveolar macrophages, and were most likely due to a contamination issue affecting that particular TEM preparation. The newly acquired TEM data confirm the expected intracellular morphology of Mtb within AMLs, as previously described (1, 2), and are fully consistent with the biological framework of the study. (lines 405-410).

With respect to the reviewer's broader concern about potential contamination affecting other experiments described in the original version of the manuscript, we would like to clarify that we found no evidence supporting this possibility. Indeed, all other functional experiments performed with AMLs (including cell viability assays, flow cytometry-based phenotype, and anti-TB susceptibility profiling of bacterial load and cytokine secretion) were conducted 6 days post-infection. In every single experiment, bacterial burden was systematically quantified by CFU plating. Across all replicates and donors, the only colonies that grew over the 3 weeks of incubation were those corresponding to Mtb. At no point did we detect additional colony morphologies suggestive of contamination, which would have grown much faster than Mtb.

Furthermore, throughout these experiments, we never observed critical signs of bacterial contamination, including:

- Acidification of the culture medium,
- Turbidity of the medium,
- Unexpected alterations in cell morphology,
- Growth of non-mycobacterial colonies in CFU assays.

Taken together, the systematic CFU monitoring, the absence of phenotypic contamination signs, and the reproducibility of the biological results across independent experiments strongly support the conclusion that the potential contamination was restricted to the specific TEM preparation originally shown, and did not impact the other datasets presented in the manuscript.

We fully agree with the reviewer that transparency is essential. For this reason, we have:

1. Repeated the TEM experiments.
2. Replaced the original images with new validated images.
3. Explicitly clarified in the revised manuscript that the TEM analysis was repeated following reviewer feedback.
4. Confirmed, with experimental evidence, that only Mtb was present in all functional experiments.

Importantly, none of the key conclusions of the manuscript are affected, as they are supported by:

- Reproducible CFU measurements,
- Flow cytometry analyses,
- Cytokine quantifications,
- Already published transcriptomic datasets,
- Independent biological replicates.

We are grateful to the reviewer for raising this concern, as it has allowed us to strengthen the rigor and clarity of the manuscript. We hope that the new data and the detailed clarification provided here satisfactorily address the raised concerns.

Regarding the use of antibiotics in cell culture, we apologize for the lack of clarity in the original manuscript. In fact, both AML and MDM cultures were differentiated in the presence of antibiotics (Penicillin–Streptomycin), which were removed from both cultures exclusively prior to Mtb infection. There was therefore no differential antibiotic exposure between AML and MDM that could have compromised sterility in one condition but not the other. We have now corrected and clarified this point in the Methods section of the revised manuscript to avoid any ambiguity. (line 164)

2.1) Fig4. Lines 508-509 - The authors report that as expect anti-TB drug are effective except PZA that does not affect replication dynamics as shown in their Heat-Map with an approximate 0% inhibition across 7 donors. Then, Lines 513-514 they state the following "All standard-of-care antibiotics, including PZA, significantly suppressed bacterial replication." So PZA does not affect replication dynamics with the Lux Reporter, but affects viability according to CFU counting assay with a 50% reduction in CFU. Can the authors comment on these discrepancies? And discuss about how reliable the Lux system is?

We agree with the reviewer that PZA shows an apparent divergence between the luminescence (Lux) readout and CFU enumeration. Importantly, the Lux signal (RLU) is a metabolic proxy and does not directly quantify colony-forming capacity. PZA is a prodrug whose activity is highly context-dependent and requires an acidic environment to be converted into its active form, which helps explain its strong *in vivo* efficacy but often limited activity in macrophage-based *in vitro* models, including MDMs (3–6).

Accordingly, in our model, PZA induced only a modest reduction in CFU (approximately 0.5 log units) compared with other antibiotics. This decrease in viable, colony-forming bacteria was not accompanied by a proportional reduction in luminescence, indicating that a fraction of bacilli

remained metabolically active and continued to express the Lux reporter despite reduced replicative fitness. These data show that PZA can impair bacterial viability without immediately suppressing detectable metabolic activity by the Lux system, underscoring the complementary nature and limitations of luminescence-based readouts relative to CFU enumeration. For these reasons, we consider Lux a robust, high-throughput kinetic readout, but one that must be interpreted alongside CFU for compounds with context-dependent or metabolic-decoupling mechanisms. This issue has now been explicitly addressed in lines 654-659 of the revised manuscript.

2.2) Have they performed a calibration standard to correlate luminescence intensity with CFU?

Before initiating the experiments, we conducted a calibration analysis to validate the relationship between bacterial load and luminescence intensity. Bacterial numbers were initially estimated by OD measurements and subsequently adjusted by serial dilutions to obtain defined bacterial concentrations, which were then correlated with the Lux-derived luminescence signal. This analysis, now provided below as Reviewer Figure 1, is consistent with previous observations reported by Armesilla-Diaz(7), further supporting the robustness of the luminescence-based quantification approach. Accordingly, luminescence intensity exhibits a strong positive correlation with bacterial number across the tested range. Of note, this calibration was performed using bacteria grown in 7H9 culture medium under non-stress conditions. Under these conditions, bacterial metabolism and Lux expression remain relatively constant, which explains the robust correlation observed between luminescence intensity and bacterial load. Molecules that induce substantial bacterial killing, such as INH, RIF, or MXF, are consistently accompanied by reductions in bacterial load, which are consistently accompanied by decreases in luminescence (Fig. 3A and B). However, when compounds induce only a modest reduction in CFU, the remaining live bacteria that experience these stress conditions can alter their metabolic state and Lux expression. Consequently, under these conditions, changes in luminescence intensity may not always directly correlate with CFU counts, as observed for PZA, LZD, and most HDTs (ASA, SMV, and Met) (Fig. 3A and B). For these reasons, all experiments were systematically performed using both readouts, the 3-week CFU enumeration serving as the gold-standard measure of bacterial viability and replication, and luminescence quantification used as a high-throughput, fast, and complementary indicator of bacterial metabolic activity.

2.3) The timepoint T0 that corresponds to the initial inoculum in CFU or Luminescence Units should also be available for the readers, to assess the replication dynamics over 6 days as done in Fig. 5H.

We thank the reviewer for this comment. The infection kinetics, including the T0 time point corresponding to the initial inoculum, were evaluated for luminescence and CFU using the *lux* strain, as well as fluorescence using the GFP-expressing strain. These data are already

included and shown in **Supplementary Figure 3D**, allowing readers to assess replication dynamics over the 6-day period.

3) Also, its absolutely not clear what concentration of drugs were used in this assay. Please provide more information about how these experiments were performed.

The reviewer is correct to point out this omission. The concentrations of all antibiotics tested, as indicated in the corresponding figure legends, are now explicitly added and clarified in the revised manuscript. In addition, we have expanded the Methods section to provide further details on how these experiments were performed, thereby improving transparency and reproducibility. (Lines: 172-175).

Concentrations:

SoC (INH = 10 μ M ; RIF = 10 μ M ; LZD = 10 μ M ; MXF = 10 μ M ; PZA = 25 μ M)

VTT (ETZ = 10 μ M ; BBH7 = 10 μ M)

HDT (ASA = 50 μ M ; IBU = 50 μ M; SMV = 10 μ M; MET = 50 μ M; DORA = 50 μ M)

(Lines: 1143-1145)

4) Of note, a thorough side-by-side comparison of Mtb replication dynamics overtime and antibiotic efficacy in hAM vs **AML vs ALI** would definitely increase the strength of this study, and showcase how AML and ALI can be complementary and suitable model for TB-Host interactions and drug discovery studies.

We are grateful to the reviewer for this insightful comment and fully agree that a direct, side-by-side comparison of Mtb replication dynamics and antibiotic efficacy in hAM, AML, and ALI models would further strengthen the study. However, the systematic use of primary hAMs for extensive kinetic and pharmacological analyses faces major practical and ethical limitations. Primary hAMs can only be obtained through bronchoalveolar lavage (BAL) from patients, an invasive clinical procedure that requires bronchoscopy, specialized medical staff, and strict ethical approval. As a result, access to hAMs is necessarily limited to small patient cohorts, and repeated sampling is not feasible. Moreover, the cellular yield from BAL varies widely across donors and clinical contexts, and is often low, which precludes large-scale or high-throughput experimentation. In practice, the number of hAMs recovered is insufficient (approximately 2×10^5 cell per patient) to perform parallel time-course experiments, multi-drug dose-response assays, or the screening of numerous compounds under standardized conditions. For our experiments, we used at least 3×10^6 AML cells.

These constraints are precisely what motivated the core objective of our study. Rather than positioning primary hAMs as a routine experimental platform, our goal was to validate a robust, scalable, and ethically accessible human model that faithfully reproduces the phenotype and functional behavior of hAMs. As demonstrated in our work, AMLs closely recapitulate primary hAMs at the phenotypic, transcriptional, and functional levels, including their immunoregulatory programming and permissiveness to Mtb infection, and can be readily generated from peripheral blood monocytes in sufficient numbers and with high reproducibility. This makes AMLs particularly suitable for kinetic analyses, comparative drug testing, and medium- to high-throughput applications that are not achievable with primary hAMs.

In addition, we would like to emphasize that a direct comparison of Mtb replication dynamics over time and antibiotic efficacy between AML and ALI models was indeed performed in the present study. Specifically, the kinetics of bacterial burden were evaluated over time in AMLs (Supplementary Fig. 3D) and in ALI cultures (Fig. 5H), demonstrating compartment-specific differences in Mtb replication behavior. Furthermore, the efficacy of standard-of-care antibiotics, host-directed therapies, and virulence-targeting compounds was systematically assessed side-by-side in AML (Fig. 4) and ALI (Fig. 6) systems. These data directly illustrate how the intracellular macrophage niche and the extracellular airway epithelial niche differentially shape

drug activity and inflammatory responses, thereby supporting the complementary value of AML and ALI models for studying TB host–pathogen interactions and therapeutic interventions.

5) Fig5 Lines 569-572 the authors state that "Strikingly, rather than replicating, Mtb showed a progressive decline over time in both luminescence and viable counts, indicating not only a lack of replication but a potential loss of bacterial viability in this model". Have the authors performed any statistics to confirm that the decrease they observed is significant? As it seems that this decrease remains minimal with close to half a Log₁₀.

We thank the reviewer for this important comment. The appropriate statistical analyses were performed to assess the temporal evolution of Mtb burden in the ALI model. These analyses confirmed that the decrease observed over time in both luminescence and CFU measurements is statistically significant. We have now corrected this point by explicitly adding statistical annotations to Fig. 5 to indicate significant differences between time points. Although the magnitude of the decrease is modest (approximately 0.5 log₁₀), it is consistent and reproducible across independent experiments and is statistically significant. Importantly, this reduction reflects a biologically meaningful phenotype in the ALI system, in which extracellular Mtb fails to replicate and progressively loses viability, rather than undergoing active intracellular growth in epithelial cells, consistent with previous reports using airway organoid models (8).

6) Fig6. All the concerns about Fig4 applies to this Figure as well (T0 CFU, Concentration of antibiotics used etc). It is essential to define whether antibiotics are simply blocking replication or have a bactericidal effect. In addition, can the authors comments about the discrepancies they have in this system with LZD which is showing 70-100% inhibition using the Mtb Lux system but no changes in term of CFU.

We agree with the reviewer. First, regarding bacterial load normalization and replication dynamics, the initial inoculum (Day 0, T0) and the final bacterial burden at Day 6 are explicitly shown in Fig. 5H, for both luminescence and CFU measurements.

Second, the concentrations of all compounds tested have now been explicitly added to the figure legend. The concentrations used were as follows: standard-of-care antibiotics (SoC: INH, RIF, LZD, and MXF at 10 μM; PZA at 25 μM), virulence-targeting therapies (VTT: ETZ and BBH7 at 10 μM), and host-directed therapies (HDT: ASA and IBU at 50 μM; SMV at 10 μM; MET and DORA at 50 μM). (Lines: 1172-1174).

Third, the apparent discrepancy observed with LZD is consistent with its well-established pharmacological mechanism of action. Briefly, LZD is a bacteriostatic antibiotic that inhibits bacterial protein synthesis by binding to the 50S ribosomal subunit and blocking the formation of the initiation complex (9, 10). As a consequence, this molecule rapidly suppresses the synthesis of bacterial proteins, including the Lux reporter, resulting in a marked reduction in luminescence. This explains the strong inhibitory effect detected using the Mtb Lux system. However, because linezolid is bacteriostatic rather than bactericidal, it primarily prevents bacterial growth and replication, rather than inducing rapid bacterial death. As a result, the number of viable, colony-forming bacteria remains largely unchanged over the experimental time frame, accounting for the limited reduction in CFU counts. This issue has now been explicitly addressed in lines 713–717 of the revised manuscript.

Finally, in our models, INH, RIF, and MXF consistently induced strong reductions in CFU counts, demonstrating a clear bactericidal effect. These decreases were accompanied by concordant reductions in luminescence, indicating both loss of bacterial viability and metabolic activity. In contrast, LZD displayed a bacteriostatic profile, as expected from its known mechanism of action discussed above. In addition, PZA exhibited a comparatively weak effect in our *in vitro* systems. This observation is consistent with extensive literature showing that PZA exhibits

limited activity in standard *in vitro* models, as its antibacterial efficacy depends on conversion to its active form in acidic environments, a condition more faithfully achieved *in vivo* than in macrophage- or epithelial-based *in vitro* cultures (4–6).

Minor Point:

Have the authors considered using an aminoglycoside in their assays with AML and ALI as control as they are known for their poor permeability within cells, but good efficacy against free extracellular bacteria?

We thank the reviewer for this suggestion and agree that including an aminoglycoside control could be informative. However, in the present study, we deliberately focused on a defined set of reference compounds selected to benchmark intracellular versus airway-associated niches using antibiotics, host-directed therapies, and virulence-targeting agents that are widely employed in current TB research and clinical practice. Aminoglycosides have played a major historical role in TB treatment and remain pharmacologically relevant as second-line injectable drugs for treatment of MDR-TB (11). However, their use has progressively declined over recent years due to toxicity concerns (mainly hearing loss and nephrotoxicity) and the availability of more effective and better-tolerated oral regimens (11–13). Consequently, they are less frequently used as reference compounds in contemporary experimental and translational TB studies.

REVIEWER 2

Major Points

1) The Introduction is thorough and well-referenced but would benefit from greater concision. The central hypothesis and specific objectives should be stated more explicitly near the end of the section. In addition, briefly clarifying how the AML and ALI models are intended to complement each other would help orient readers before the Results.

We thank the reviewer for these helpful suggestions. In response, we have revised the Introduction to improve clarity and concision by refining the final section to more explicitly state the study's central hypothesis and specific objectives (lines 124-127). In addition, we have clarified how the AML and ALI models are intended to complement each other, with the AML model enabling detailed analysis of macrophage heterogeneity and host–pathogen interactions, and the ALI model providing a physiologically relevant airway context to assess antimicrobial activity and drug responses at the epithelial interface (lines 130-134).

2) The AML model is convincingly validated, and the identification of infection-associated heterogeneity within AMLs is a notable strength. However, several Results sections are overly detailed, particularly those describing transcriptomic analyses, and could be streamlined. The interpretation of discrepancies between luminescence-based readouts and CFU measurements should be clarified. It would also be helpful to more clearly state whether the identified AML subsets represent stable populations or dynamic infection-induced states.

We appreciate this thoughtful and constructive comment. In response, we have streamlined Results sections, particularly those describing transcriptomic analyses, by focusing on the key findings.

We have also clarified the interpretation of discrepancies between luminescence-based readouts and CFU measurements, explicitly discussing the metabolic nature of luminescence signals and their limitations for distinguishing between growth inhibition and bacterial killing (Lines 654-659; 676-681; and 713-717). Finally, we have revised the Discussion to more clearly

state that the identified AML subsets are best interpreted as dynamic activation states associated with infection, rather than as fixed or ontogenetically distinct macrophage populations (Lines: 633-635). These clarifications strengthen the study's conceptual framework without altering its main conclusions.

3) The Discussion effectively integrates the findings and places them in the context of current TB literature.

We thank the reviewer for this positive assessment.

4) That said, conclusions about host-directed therapies should be made more cautiously, since immunomodulatory effects do not always indicate therapeutic benefit. The model limitations, including the lack of adaptive immune components, could be more openly acknowledged.

We agree with the reviewer that immunomodulatory effects do not necessarily translate into therapeutic benefit and should therefore be interpreted cautiously. In the revised manuscript, we have tempered our conclusions regarding host-directed therapies and explicitly acknowledge the limitations of the *in vitro* models used, including the absence of adaptive immune components. We now emphasize that modulation of pro-inflammatory cytokine production was observed in a human *in vitro* model that mimics the lung microenvironment, which represents the main site of infection; it should be viewed not as a replacement but as a complementary approach that may help bridge responses observed in preclinical models to those in humans. Nevertheless, we stress that *in vivo* studies remain essential to determine whether these immunomodulatory profiles are preserved in a physiological context that allows interactions with adaptive immune responses and other systemic and tissue-level regulatory components. (lines 775-782).

5) While the results are clearly presented, the section could be strengthened by placing more emphasis on conceptual synthesis to integrate the findings, rather than restating individual observations.

We thank the reviewer for this insightful comment. In response, we have revised the Results section to place greater emphasis on conceptual synthesis and integration of the findings. Specifically, we have added integrative statements at the end of key sections to clearly articulate how individual observations collectively support the overarching framework of niche-specific host-pathogen interactions and differential drug responses. (lines 561-563; 590-595).

6) A significant limitation is that most conclusions remain superficial. The study catalogs differences in bacterial load and cytokine production across models but offers little mechanistic explanation for these differences. For example, pyrazinamide activity in AMLs but not in ALI cultures is interpreted as niche-specific efficacy, but no direct evidence is provided regarding intracellular pH, drug activation, or bacterial metabolic state in AMLs versus epithelial surfaces. Further explanation would help.

We thank the reviewer for this thoughtful comment and agree that a deeper mechanistic dissection of the niche-dependent differences observed between AML and ALI models would further enrich the study. Our primary objective, however, was to establish and benchmark a dual human lung platform capable of capturing compartment-specific differences in drug efficacy and host response, rather than to fully resolve the underlying molecular mechanisms governing these differences.

Regarding pyrazinamide (PZA), its activity is well known to depend on environmental factors such as acidic pH and bacterial metabolic state, as it requires conversion to pyrazinoic acid by the bacterial enzyme PncA and exhibits enhanced activity under acidic conditions that promote weak-acid permeation and intrabacterial pH disruption (4-6). The selective activity observed in AMLs, but not in ALI cultures, is therefore biologically consistent with the distinct

microenvironments represented by these systems. AMLs recapitulate an intracellular macrophage niche characterized by phagosomal acidification, metabolic stress, and host-derived pressures that are known to potentiate PZA activity. In contrast, the ALI model represents an extracellular airway environment, where bacteria remain apically localized, and are unlikely to experience comparable intracellular acidification or stress signals. Notably, treatment of AMLs with pyrazinoic acid (POA), the active metabolite of PZA, resulted in a modest but statistically significant reduction in CFU of approximately 50% relative to untreated controls (Reviewer Fig. 2), similar to the antibacterial effect observed with PZA (Fig. 4b). These results indicate that administration of the active compound did not further enhance bacterial killing in this system, and suggest that PZA is properly metabolized as an active compound in the AMLs. We agree that direct measurements of intracellular pH, PZA activation dynamics, or bacterial metabolic state would provide mechanistic confirmation of this interpretation. However, such investigations would require dedicated biochemical and bacterial reporter approaches that fall beyond the scope of the present study, whose the central aim is to establish a physiologically relevant and scalable comparative platform for preclinical drug evaluation. To clarify this point for the reader, we have revised the Discussion section to explicitly address the mechanistic plausibility of the observed niche-dependent PZA activity, while clearly acknowledging that direct experimental validation of these mechanisms, including comparative analyses between AML and MDM models, will require dedicated future investigations. (lines 662-668).

7) CD14/CD16-based stratification is informative, but the relationship to bona fide resident versus recruited AM populations *in vivo* remains speculative.

We thank the reviewer for raising this important point. We agree that CD14/CD16-based stratification does not allow definitive discrimination between bona fide resident and recruited alveolar macrophage populations *in vivo*. Importantly, our intention was not to assign ontogenetic identity to the CD14⁺CD16⁺ or CD14⁺CD16⁻ AML subsets, but rather to describe distinct activation and functional states within an alveolar macrophage-like compartment.

To avoid overinterpretation, we have revised the Discussion to explicitly clarify that CD14/CD16 expression is used here as a functional and phenotypic stratification that may relate to differential permissiveness to *Mtb* infection, but does not imply resident versus recruited macrophage identity *in vivo*. (Lines: 633-635).

8) The manuscript acknowledges this issue but still draws conclusions about bacteriostatic versus bactericidal activity, particularly for linezolid.

We agree with the reviewer that distinguishing bacteriostatic from bactericidal activity requires careful interpretation, particularly when relying on metabolic readouts. In the revised manuscript, we have therefore refined our wording to avoid overinterpretation and to more explicitly anchor our conclusions in the CFU-based measurements.

Specifically, for linezolid, we now clarify that the marked reduction in luminescence is consistent with its well-established bacteriostatic mechanism of action, which involves inhibition of bacterial protein synthesis through binding to the 50S ribosomal subunit. As luminescence depends on active protein translation, inhibition of ribosomal function is expected to strongly reduce the metabolic signal without necessarily affecting bacterial viability, in line with the unchanged CFU counts. These revisions ensure that our conclusions accurately reflect the acknowledged methodological limitations while preserving the biological interpretation of the data. (Lines: 713-717).

9) Cytokine measurements (TNF- α , IL-6, IL-8) are used as proxies for disease severity, but the biological implications are not always clear. Reduced cytokine production is implicitly framed as beneficial, yet excessive suppression of TNF or IL-6 could also impair bacterial control *in vivo*. We appreciate this important and nuanced point. We agree that cytokine levels should be interpreted as readouts of immune modulation rather than direct surrogates of therapeutic benefit or disease outcome. However, we would like to emphasize that metformin and doramapimod, both used to benchmark the dual human platform, and both reducing pro-inflammatory cytokines, are already known to be efficient *in vivo* at reducing bacterial load, which is associated with reduced pro-inflammatory response and immunopathology (14, 15), supporting that our dual platform are indicative of key parameters of compound efficacy and immunomodulating properties. In the revised manuscript, we have clarified that reduced cytokine production is not inherently beneficial and that excessive suppression of key innate immune mediators such as TNF- α or IL-6 could impair bacterial control *in vivo*. We now explicitly frame cytokine measurements as relevant indicators of host response modulation within each model, and emphasize that these inflammatory profiles in human settings display potential predictive value for compound down selection, and further validation of biological and therapeutic implications *in vivo*. Accordingly, we propose that this platform should be viewed as a complementary preclinical approach that may help bridge findings from diverse experimental models to humans, rather than as a replacement for *in vivo* studies. (Lines 761 – 767).

Minor Point:

1) The statistical methods should clarify how multiple comparisons were handled across the large drug panels.

In our study, all statistical analyses were based on pre-specified pairwise comparisons between each drug-treated condition and the corresponding infected, untreated control. No direct statistical comparisons between different drugs were performed, as this was not the study's objective. Therefore, a global correction for multiple comparisons was not applied. We have now clarified this point in the Statistical Analysis section of the Methods. (Line 330-332)

2) *Mycobacterium tuberculosis* should be italicized throughout the manuscript.

Mycobacterium tuberculosis is now italicized throughout the manuscript.

3) An asterisk (*) is missing in the Figure 1 legend.

Response: The missing asterisk has been added to the Figure 1 legend. (line 1105).

4) Reporting bacterial burden as CFU per well may be misleading; for in vitro assays, CFU is conventionally expressed as CFU/mL.

Bacterial burden is now reported as CFU/mL throughout the manuscript, as requested.

5) Cytokine concentrations should be reported using a consistent unit (either pg/mL or ng/mL) across all figures and text.

Cytokine concentrations are now consistently reported as pg/mL across all figures.

REFERENCES

1. Pahari S, Arnett E, Simper J, Azad A, Guerrero-Arguero I, Ye C, Zhang H, Cai H, Wang Y, Lai Z, Jarvis N, Lumbreras M, Maselli DJ, Peters J, Torrelles JB, Martinez-Sobrido L, Schlesinger LS. 2023. A new tractable method for generating human alveolar macrophage-like cells in vitro to study lung inflammatory processes and diseases. *mBio* 0:e00834-23.
2. Pahari S, Neehus A-L, Trapnell BC, Bustamante J, Casanova J-L, Schlesinger LS. 2024. Protocol to develop human alveolar macrophage-like cells from mononuclear cells or purified monocytes for use in respiratory biology research. *STAR Protoc* 5:103061.
3. Malone L, Schurr A, Lindh H, McKENZIE D, Kiser JS, Williams JH. 1952. The effect of pyrazinamide (aldinamide) on experimental tuberculosis in mice. *Am Rev Tuberc* 65:511–518.
4. Santucci P, Greenwood DJ, Fearn A, Chen K, Jiang H, Gutierrez MG. 2021. Intracellular localisation of *Mycobacterium tuberculosis* affects efficacy of the antibiotic pyrazinamide. *Nat Commun* 12:3816.
5. Santucci P, Aylan B, Botella L, Bernard EM, Bussi C, Pellegrino E, Athanasiadi N, Gutierrez MG. 2022. Visualizing Pyrazinamide Action by Live Single-Cell Imaging of Phagosome Acidification and *Mycobacterium tuberculosis* pH Homeostasis. *mBio* 13:e0011722.
6. Laudouze J, Rokitskaya TI, Abolet A, Point V, Firsov AM, Khailova LS, Cavalier J-F, Canaan S, Baulard AR, Antonenko YN, Gouzy A, Santucci P. 2025. Pyrazinamide kills *Mycobacterium tuberculosis* via pH-driven weak-acid permeation and cytosolic acidification. *bioRxiv* 2025.09.26.678883.
7. Armesilla-Diaz A, Pilar Arenaz M, Ashby C, Blanco D, D'Oria E, Garuti H, Gómez V, González-Del-Río R, Martínez-Hoyos M, Meiler E, Mendoza-Losana A, Mohamet L, Padrón-Barthe L, Pérez E, Pérez L, Remuiñán MJ, Rodríguez-Miquel B, Segura-Carro D, Viera-Morilla S. 2025. High-throughput screening

of small molecules targeting *Mycobacterium tuberculosis* in human iPSC macrophages. *Antimicrob Agents Chemother* 69:e0161324.

8. Iakobachvili N, Leon-Icaza SA, Knoops K, Sachs N, Mazères S, Simeone R, Peixoto A, Bernard C, Murriss-Espin M, Mazières J, Cam K, Chalut C, Guilhot C, López-Iglesias C, Ravelli RBG, Neyrolles O, Meunier E, Lugo-Villarino G, Clevers H, Cougoule C, Peters PJ. 2022. Mycobacteria–host interactions in human bronchiolar airway organoids. *Molecular Microbiology* 117:682–692.
9. Dietze R, Hadad DJ, McGee B, Molino LPD, Maciel ELN, Peloquin CA, Johnson DF, Debanne SM, Eisenach K, Boom WH, Palaci M, Johnson JL. 2008. Early and extended early bactericidal activity of linezolid in pulmonary tuberculosis. *Am J Respir Crit Care Med* 178:1180–1185.
10. Gan WC, Ng HF, Ngeow YF. 2023. Mechanisms of Linezolid Resistance in Mycobacteria. *Pharmaceuticals (Basel)* 16:784.
11. Frequently Asked Questions on the WHO Rapid Communication: key changes to the treatment of multidrug- and rifampicin-resistant TB. <https://www.who.int/publications/m/item/WHO-CDS-TB-2018.18>. Retrieved 20 January 2026.
12. Dillard LK, Martinez RX, Perez LL, Fullerton AM, Chadha S, McMahon CM. 2021. Prevalence of aminoglycoside-induced hearing loss in drug-resistant tuberculosis patients: A systematic review. *J Infect* 83:27–36.
13. Owusu E, Amartey BT, Afutu E, Bofofo N. 2022. Aminoglycoside Therapy for Tuberculosis: Evidence for Ototoxicity among Tuberculosis Patients in Ghana. *Diseases* 10:10.
14. Singhal A, Jie L, Kumar P, Hong GS, Leow MK-S, Paleja B, Tsenova L, Kurepina N, Chen J, Zolezzi F, Kreiswirth B, Poidinger M, Chee C, Kaplan G, Wang YT, De Libero G. 2014. Metformin as adjunct antituberculosis therapy. *Sci Transl Med* 6:263ra159.
15. Hölscher C, Gräß J, Hölscher A, Müller AL, Schäfer SC, Rybniker J. 2020. Chemical p38 MAP kinase inhibition constrains tissue inflammation and improves antibiotic activity in *Mycobacterium tuberculosis*-infected mice. *Sci Rep* 10:13629.

Re: Spectrum03729-25R1 (Dual human lung models reveal compartment-specific activity of anti-tuberculosis drugs and host-directed therapies)

Dear Dr. Caio Cesar Barbosa Bomfim:

Your manuscript has been accepted, and I am forwarding it to the ASM production staff for publication. Your paper will first be checked to make sure all elements meet the technical requirements. ASM staff will contact you if anything needs to be revised before copyediting and production can begin. Otherwise, you will be notified when your proofs are ready to be viewed.

Sincerely,
Sophia Georghiou
Editor
Microbiology Spectrum

Reviewer #1 (Comments for the Author):

I would like to thank the authors for their careful inspection of the manuscript. They have carefully revised the manuscript and addressed my comments. I have no further comment, and would like to congratulate them for this valuable piece of work.

Reviewer #2 (Comments for the Author):

The authors have adequately addressed the reviewer's suggestions in the revised manuscript. I have no further comments or requests for clarification.

Spectrum03729-25R1

I would like to thank the authors for their careful inspection of the manuscript. They have carefully revised the manuscript and adressed my comments. I have no further comment, and would like to congratulate them for this valuable piece of work.